# Higher-Order Spectral Clustering of Directed Graphs

**Steinar Laenen** *
Oxford Research Group
FiveAI
steinar.laenen@five.ai

**He Sun**
School of Informatics
University of Edinburgh
h.sun@ed.ac.uk

## Abstract

Clustering is an important topic in algorithms, and has a number of applications in machine learning, computer vision, statistics, and several other research disciplines. Traditional objectives of graph clustering are to find clusters with low conductance. Not only are these objectives just applicable for undirected graphs, they are also incapable to take the relationships between clusters into account, which could be crucial for many applications. To overcome these downsides, we study directed graphs (digraphs) whose clusters exhibit further "structural" information amongst each other. Based on the Hermitian matrix representation of digraphs, we present a nearly-linear time algorithm for digraph clustering, and further show that our proposed algorithm can be implemented in sublinear time under reasonable assumptions. The significance of our theoretical work is demonstrated by extensive experimental results on the UN Comtrade Dataset: the output clustering of our algorithm exhibits not only how the clusters (sets of countries) relate to each other with respect to their import and export records, but also how these clusters evolve over time, in accordance with known facts in international trade.

## 1 Introduction

Clustering is one of the most fundamental problems in algorithms and has applications in many research fields including machine learning, network analysis, and statistics. Data can often be represented by a graph (e.g., users in a social network, servers in a communication network), and this makes graph clustering a natural choice to analyse these datasets. Over the past three decades, most studies on undirected graph clustering have focused on the task of partitioning with respect to the edge densities, i.e., vertices form a cluster if they are better connected to each other than to the rest of the graph. The well-known normalised cut value [25] and graph conductance [20] capture these classical definitions of clusters, and have become the objective functions of most undirected graph clustering algorithms.

While the design of these algorithms has received a lot of research attention from both theoretical and applied research areas, these algorithms are usually unable to uncover *higher-order structural* information among clusters in *directed graphs* (digraphs). For example, let us look at the international oil trade network [29], which employs digraphs to represent how mineral fuels and oils are imported and exported between countries. Although this highly connected digraph presents little cluster structure with respect to a typical objective function of undirected graph clustering, from an economic point of view this digraph clearly exhibits a structure of clusters: there is a cluster of countries mainly exporting oil, a cluster mainly importing oil, and several clusters in the middle of this trade chain. All these clusters are characterised by the imbalance of the edge directions between clusters, and further present a clear ordering reflecting the overall trade pattern. This type of structure is not only found in trade data, but also in many other types of data such as migration data and infectious disease

spreading data. We view these types of patterns as a *higher-order* structure among the clusters and, in our point of view, this structural information could be as important as the individual clusters themselves.

**Our contribution.** In this work we study clustering algorithms for digraphs whose cluster structure is defined with respect to the imbalance of edge densities as well as the edge directions between the clusters. Formally, for any set of vertices $S_0, \ldots, S_{k-1}$ that forms a partition of the vertex set $V(G)$ of a digraph $G$, we define the *flow ratio* of $\{S_j\}_{j=0}^{k-1}$ by

$$\sum_{j=1}^{k-1} \frac{w(S_j, S_{j-1})}{\mathrm{vol}(S_j) + \mathrm{vol}(S_{j-1})},$$

where $w(S, T) \triangleq \sum_{\substack{(u,v) \in E \\ u \in S, v \in T}} w(u, v)$ is the cut value from $S \subset V$ to $T \subset V$ and $\mathrm{vol}(S)$ is the sum of degrees of the vertices in $S$. We say that $\{S_j\}_{j=0}^{k-1}$ forms an optimal partition if this $\{S_j\}_{j=0}^{k-1}$ *maximises* the flow ratio over all possible partitions. By introducing a complex-valued representation of the graph Laplacian matrix $\mathcal{L}_G$, we show that this optimal partition $\{S_j\}_{j=0}^{k-1}$ is well embedded into the bottom eigenspace of $\mathcal{L}_G$. To further exploit this novel and intriguing connection, we show that an approximate partition with bounded approximation guarantee can be computed in time nearly-linear in the number of edges of the input graph. In the settings for which the degrees of the vertices are known in advance, we also present a sub-linear time implementation of the algorithm. The significance of our work is further demonstrated by experimental results on several synthetic and real-world datasets. In particular, on the UN Comtrade dataset our clustering results are well supported by the literature from other research fields. At the technical level, our analysis could be viewed as a hybrid between the proof of the Cheeger inequality [6] and the analysis of spectral clustering for undirected graphs [23], as well as a sequence of recent work on fast constructions of graph sparsification (e.g., [26]). We believe our analysis for the new Hermitian Laplacian $\mathcal{L}_G$ could inspire future research on studying the clusters' higher-order structure using spectral methods.

**Related work.** There is a rich literature on spectral algorithms for graph clustering. For undirected graph clustering, the works most related to ours are [23, 25, 30]. For digraph clustering, [24] proposes to perform spectral clustering on the symmetrised matrix $A = M^{\mathsf{T}}M + MM^{\mathsf{T}}$ of the input graph's adjacency matrix $M$; [9] initiates the studies of spectral clustering on complex-valued Hermitian matrix representations of digraphs, however their theoretical analysis only holds for digraphs generated from the stochastic block model. Our work is also linked to analysing higher-order structures of clusters in undirected graphs [4, 5, 31], and community detection in digraphs [7, 21]. The main takeaway is that there is no previous work which analyses digraph spectral clustering algorithms to uncover the higher-order structure of clusters in a general digraph.

## 2 Preliminaries

Throughout the paper, we always assume that $G = (V, E, w)$ is a digraph with $n$ vertices, $m$ edges, and weight function $w : V \times V \to \mathbb{R}_{\geqslant 0}$. We write $u \rightsquigarrow v$ if there is an edge from $u$ to $v$ in the graph. For any vertex $u$, the in-degree and out-degree of $u$ are defined as $d_u^{\mathrm{in}} \triangleq \sum_{v:v \rightsquigarrow u} w(v, u)$ and $d_u^{\mathrm{out}} \triangleq \sum_{v:u \rightsquigarrow v} w(u, v)$, respectively. We further define the total degree of $u$ by $d_u \triangleq d_u^{\mathrm{in}} + d_u^{\mathrm{out}}$, and define $\mathrm{vol}(S) \triangleq \sum_{u \in S} d_u$ for any $S \subseteq V$. For any set of vertices $S$ and $T$, the symmetric difference between $S$ and $T$ is defined by $S \triangle T \triangleq (S \setminus T) \cup (T \setminus S)$.

Given any digraph $G$ as input, we use $M \in \mathbb{R}^{n \times n}$ to denote the adjacency matrix of $G$, where $M_{u,v} = w(u, v)$ if there is an edge $u \rightsquigarrow v$, and $M_{u,v} = 0$ otherwise. We use $A \in \mathbb{C}^{n \times n}$ to represent the Hermitian adjacency matrix of $G$, where $A_{u,v} = \overline{A_{v,u}} = w(u, v) \cdot \omega_{\lceil 2\pi k \rceil}$ if $u \rightsquigarrow v$, and $A_{u,v} = 0$ otherwise. Here, $\omega_{\lceil 2\pi k \rceil}$ is the $\lceil 2\pi k \rceil$-th root of unity, and $\overline{x}$ is the conjugate of $x$. The normalised Laplacian matrix of $G$ is defined by $\mathcal{L}_G \triangleq I - D^{-1/2} A D^{-1/2}$, where the degree matrix $D \in \mathbb{R}^{n \times n}$ is defined by $D_{u,u} = d_u$, and $D_{u,v} = 0$ for any $u \neq v$. We sometimes drop the subscript $G$ if the underlying graph is clear from the context.

For any Hermitian matrix $A \in \mathbb{C}^{n \times n}$ and non-zero vector $x \in \mathbb{C}^n$, the Rayleigh quotient $\mathcal{R}(A, x)$ is defined as $\mathcal{R}(A, x) \triangleq x^* A x / x^* x$, where $x^*$ is the complex conjugate transpose of $x \in \mathbb{C}^n$. For any

Hermitian matrix $B \in \mathbb{C}^{n \times n}$, let $\lambda_1(B) \leqslant \ldots \leqslant \lambda_n(B)$ be the eigenvalues of $B$ with corresponding eigenvectors $f_1, \ldots, f_n$, where $f_j \in \mathbb{C}^n$ for any $1 \leqslant j \leqslant n$.

## 3 Encoding the flow-structure into $\mathcal{L}_G$'s bottom eigenspace

Now we study the structure of clusters with respect to their flow imbalance, and their relation to the bottom eigenspace of the normalised Hermitian Laplacian matrix. For any set of vertices $S_0, \ldots, S_{k-1}$, we say that $S_0, \ldots, S_{k-1}$ form a $k$-way partition of $V(G)$, if it holds that $\bigcup_{0 \leqslant j \leqslant k-1} S_j = V(G)$ and $S_j \cap S_\ell = \emptyset$ for any $j \neq \ell$. As discussed in Section 1, the primary focus of the paper is to study digraphs in which there are significant connections from $S_j$ to $S_{j-1}$ for any $1 \leqslant j \leqslant k - 1$. To formalise this, we introduce the notion of *flow ratio* of $\{S_j\}_{j=0}^{k-1}$, which is defined by

$$\Phi_G(S_0, \ldots, S_{k-1}) \triangleq \sum_{j=1}^{k-1} \frac{w(S_j, S_{j-1})}{\mathrm{vol}(S_j) + \mathrm{vol}(S_{j-1})}. \tag{1}$$

We call this $k$-way partition $\{S_j\}$ an *optimal clustering* if the flow ratio given by $\{S_j\}$ achieves the maximum defined by

$$\theta_k(G) \triangleq \max_{\substack{S_0, \ldots, S_{k-1} \\ \cup S_i = V, S_j \cap S_\ell = \emptyset}} \Phi_G(S_0, \ldots, S_{k-1}). \tag{2}$$

Notice that, for any two consecutive clusters $S_j$ and $S_{j-1}$, the value $w(S_j, S_{j-1}) \cdot (\mathrm{vol}(S_j) + \mathrm{vol}(S_{j-1}))^{-1}$ evaluates the ratio of the total edge weight in the cut $(S_j, S_{j-1})$ to the total weight of the edges with endpoints in $S_j$ or $S_{j-1}$; moreover, only $k - 1$ out of $2 \cdot \binom{k}{2}$ different cuts among $S_0, \ldots, S_{k-1}$ contribute to $\Phi_G(S_0, \ldots, S_{k-1})$ according to (1). We remark that, although the definition of $\Phi_G(S_0, \ldots, S_{k-1})$ shares some similarity with the normalised cut value for undirected graph clustering [25], in our setting an optimal clustering is the one that *maximises* the flow ratio. This is in a sharp contrast to most objective functions for undirected graph clustering, whose aim is to find clusters of low conductance[2]. In addition, it is not difficult to show that this problem is NP-hard since, when $k = 2$, our problem is exactly the MAX DICUT problem studied in [15].

To study the relationship between the flow structure among $S_0, \ldots, S_{k-1}$ and the eigen-structure of the normalised Laplacian matrix of the graph, we define for every optimal cluster $S_j$ $(0 \leqslant j \leqslant k-1)$ an indicator vector $\chi_j \in \mathbb{C}^n$ by $\chi_j(u) \triangleq (w_{\lceil 2\pi \cdot k \rceil})^j$ if $u \in S_j$ and $\chi_j(u) = 0$ otherwise. We further define the normalised indicator vector of $\chi_j$ by

$$\widehat{\chi_j} \triangleq \frac{D^{1/2} \chi_j}{\|D^{1/2} \chi_j\|},$$

and set

$$y \triangleq \frac{1}{\sqrt{k}} \sum_{j=0}^{k-1} \widehat{\chi_j}. \tag{3}$$

We highlight that, due to the use of complex numbers, a single vector $y$ is sufficient to encode the structure of $k$ clusters: this is quite different from the case of undirected graphs, where $k$ mutually perpendicular vectors are needed in order to study the eigen-structure of graph Laplacian and the cluster structure [20, 23, 30]. In addition, by the use of roots of unity in (3), different clusters are separated from each other by angles, indicating that the use of a single eigenvector could be sufficient to approximately recover $k$ clusters. Our result on the relationship between $\lambda_1(\mathcal{L}_G)$ and $\theta_k(G)$ is summarised as follows:

**Lemma 3.1.** *Let $G = (V, E, w)$ be a weighted digraph with normalised Hermitian Laplacian $\mathcal{L}_G \in \mathbb{C}^{n \times n}$. Then, it holds that $\lambda_1(\mathcal{L}_G) \leqslant 1 - \frac{4}{k} \cdot \theta_k(G)$. Moreover, $\theta_k(G) = k/4$ if $G$ is a bipartite digraph with all the edges having the same direction, and $\theta_k(G) < k/4$ otherwise.*

Notice that the bipartite graph $G$ with $\theta_k(G) = k/4$ is a trivial case for our problem; hence, without lose of generality we assume $\theta_k(G) < k/4$ in the following analysis. To study how the distribution of eigenvalues influences the cluster structure, similar to the case of undirected graphs we introduce the parameter $\gamma$ defined by

$$\gamma_k(G) \triangleq \frac{\lambda_2}{1 - (4/k) \cdot \theta_k(G)}.$$

Our next theorem shows that the structure of clusters in $G$ and the eigenvector corresponding to $\lambda_1(\mathcal{L}_G)$ can be approximated by each other with approximation ratio inversely proportional to $\gamma_k(G)$.

**Theorem 3.2.** *The following statements hold: (1) there is some $\alpha \in \mathbb{C}$ such that the vector $\widetilde{f}_1 = \alpha f_1$ satisfies $\|y - \widetilde{f}_1\|^2 \leqslant 1/\gamma_k(G)$; (2) there is some $\beta \in \mathbb{C}$ such that the vector $\widetilde{y} = \beta y$ satisfies $\|f_1 - \widetilde{y}\|^2 \leqslant 1/(\gamma_k(G) - 1)$.*

# 4   Algorithm

In this section we discuss the algorithmic contribution of the paper. In Section 4.1 we will describe the main algorithm, and its efficient implementation based on nearly-linear time Laplacian solvers; we will further present a sub-linear time implementation of our algorithm, assuming the degrees of the vertices are known in advance. The main technical ideas used in analysing the algorithms will be discussed in Section 4.2.

## 4.1   Algorithm Description

**Main algorithm.**   We have seen from Section 3 that the structure of clusters is approximately encoded in the bottom eigenvector of $\mathcal{L}_G$. To exploit this fact, we propose to embed the vertices of $G$ into $\mathbb{R}^2$ based on the bottom eigenvector of $\mathcal{L}_G$, and apply $k$-means on the embedded points. Our algorithm, which we call `SimpleHerm`, only consists of a few lines of code and is described as follows: (1) compute the bottom eigenvector $f_1 \in \mathbb{C}^n$ of the normalised Hermitian Laplacian matrix $\mathcal{L}_G$ of $G$; (2) compute the embedding $\{F(v)\}_{v \in V[G]}$, where $F(v) \triangleq \frac{1}{\sqrt{d_v}} \cdot f_1(v)$ for any vertex $v$; (3) apply $k$-means on the embedded points $\{F(v)\}_{v \in V[G]}$.

We remark that, although the entries of $\mathcal{L}_G$ are complex-valued, some variant of the graph Laplacian solvers could still be applied for our setting. For most practical instances, we have $k = O(\log^c n)$ for some constant $c$, in which regime our proposed algorithm runs in nearly-linear time[3]. We refer a reader to [22] on technical discussion on the algorithm of approximating $f_1$ in nearly-linear time.

**Speeding-up the runtime of the algorithm.**   Since $\Omega(m)$ time is needed for any algorithm to read an entire graph, the runtime of our proposed algorithm is optimal up to a poly-logarithmic factor. However we will show that, when the vertices' degrees are available in advance, the following sub-linear time algorithm could be applied before the execution of the main algorithm, and this will result in the algorithm's total runtime to be sub-linear in $m$.

More formally, our proposed sub-linear time implementation is to construct a sparse subgraph $H$ of the original input graph $G$, and run the main algorithm on $H$ instead. The algorithm for obtaining graph $H$ works as follows: every vertex $u$ in the graph $G$ checks each of its outgoing edges $e = (u, v)$, and samples each outgoing edge with probability

$$p_u(u, v) \triangleq \min\left\{w(u, v) \cdot \frac{\alpha \cdot \log n}{\lambda_2 \cdot d_u^{\mathrm{out}}}, 1\right\};$$

in the same time, every vertex $v$ checks each of its incoming edges $e = (u, v)$ with probability

$$p_v(u, v) \triangleq \min\left\{w(u, v) \cdot \frac{\alpha \cdot \log n}{\lambda_2 \cdot d_v^{\mathrm{in}}}, 1\right\},$$

where $\alpha \in \mathbb{R}_{>0}$ is some constant which can be determined experimentally. As the algorithm goes through each vertex, it maintains all the sampled edges in a set $F$. Once all the edges have

been checked, the algorithm returns a weighted graph $H = (V, F, w_H)$, where each sampled edge $e = (u, v)$ has a new weight defined by $w_H(u, v) = w(u, v)/p_e$. Here, $p_e$ is the probability that $e$ is sampled by one of its endpoints and, for any $e = (u, v)$, we can write $p_e$ as $p_e = p_u(u, v) + p_v(u, v) - p_u(u, v)p_v(u, v)$.

## 4.2  Analysis

**Analysis of the main algorithm.**   Now we analyse the proposed algorithm, and prove that running $k$-means on $\{F(v)\}_{v \in V[G]}$ is sufficient to obtain a meaningful clustering with bounded approximation guarantee. We assume that the output of a $k$-means algorithm is $A_0, \ldots, A_{k-1}$. We define the cost function of the output clustering $A_0, \ldots, A_{k-1}$ by

$$\mathsf{COST}(A_0, \ldots, A_{k-1}) \triangleq \min_{c_0, \ldots, c_{k-1} \in \mathbb{C}} \sum_{j=0}^{k-1} \sum_{u \in A_j} d_u \|F(u) - c_j\|^2,$$

and define the optimal clustering by

$$\Delta_k^2 \triangleq \min_{\text{partition } A_0, \ldots A_{k-1}} \mathsf{COST}(A_0, \ldots, A_{k-1}).$$

Although computing the optimal clustering for $k$-means is NP-hard, we will show that the cost value for the optimal clustering can be upper bounded with respect to $\gamma_k(G)$. To achieve this, we define $k$ points $p^{(0)}, \ldots, p^{(k-1)}$ in $\mathbb{C}$, where $p^{(j)}$ is defined by

$$p^{(j)} = \frac{\beta}{\sqrt{k}} \cdot \frac{(\omega_{\lceil 2\pi \cdot k \rceil})^j}{\sqrt{\mathrm{vol}(S_j)}}, \qquad 0 \leqslant j \leqslant k - 1. \tag{4}$$

We could view these $p^{(0)}, \ldots, p^{(k-1)}$ as approximate centers of the $k$ clusters, which are separated from each other through different powers of $\omega_{\lceil 2\pi \cdot k \rceil}$.

Our first lemma shows that the total distance between the embedded points from every $S_j$ and their respective centers $p^{(j)}$ can be upper bounded, which is summarised as follows:

**Lemma 4.1.** *It holds that* $\sum_{j=0}^{k-1} \sum_{u \in S_j} d_u \cdot \left\| F(u) - p^{(j)} \right\|^2 \leqslant (\gamma_k(G) - 1)^{-1}$.

Since the cost value of the optimal clustering is the minimum over all possible partitions of the embedded points, by Lemma 4.1 we have that $\Delta_k^2 \leqslant (\gamma_k(G) - 1)^{-1}$. We assume that the $k$-means algorithm used here achieves an approximation ratio of $\mathsf{APT}$. Therefore, the output $A_0, \ldots, A_{k-1}$ of this $k$-means algorithm satisfies $\mathsf{COST}(A_0, \ldots, A_{k-1}) \leqslant \mathsf{APT}/(\gamma_k(G) - 1)$.

Secondly, we show that the norm of the approximate centre of each cluster is *inversely* proportional to the volume of each cluster. This implies that larger clusters are closer to the origin, while smaller clusters are further away from the origin.

**Lemma 4.2.** *It holds for any* $0 \leqslant j \leqslant k - 1$ *that* $\left\| p^{(j)} \right\|^2 = \|\beta\|^2 \cdot (k \cdot \mathrm{vol}(S_j))^{-1}$.

Thirdly, we prove that the distance between different approximate centres $p^{(j)}$ and $p^{(\ell)}$ is inversely proportional to the volume of the *smaller* cluster, which implies that the embedded points of the vertices from a smaller cluster are far from the embedded points from other clusters. This key fact explains why our algorithm is able to approximately recover the structure of all the clusters.

**Lemma 4.3.** *It holds for any* $0 \leqslant j \neq \ell \leqslant k - 1$ *that* $\left\| p^{(j)} - p^{(\ell)} \right\|^2 \geqslant \frac{\|\beta\|^2}{3k^3 \cdot \min\{\mathrm{vol}(S_j), \mathrm{vol}(S_\ell)\}}$.

Combining these three lemmas with some combinatorial analysis, we prove that the symmetric difference between every returned cluster by the algorithm and its corresponding cluster in the optimal partition can be upper bounded, since otherwise the cost value of the returned clusters would contradict Lemma 4.1.

**Theorem 4.4.** *Let* $G = (V, E)$ *be a digraph, and* $S_0, \ldots, S_{k-1}$ *be a $k$-way partition of $V[G]$ that maximises the flow ratio* $\Phi_G(S_0, \ldots, S_{k-1})$. *Then, there is an algorithm that returns a $k$-way partition $A_0, \ldots, A_{k-1}$ of $V[G]$. Moreover, by assuming $A_j$ corresponds to $S_j$ in the optimal partition, it holds that* $\mathrm{vol}(A_j \triangle S_j) \leqslant \varepsilon \mathrm{vol}(S_j)$ *for some* $\varepsilon = 48k^3 \cdot (1 + \mathsf{APT}) / (\gamma_k(G) - 1) \leqslant 1/2$.

We remark that the analysis of our algorithm is similar with the work of [23]. However, the analysis in [23] relies on $k$ indicator vectors of $k$ clusters, each of which is in a different dimension of $\mathbb{R}^n$; this implies that $k$ eigenvectors are needed in order to find a good $k$-way partition. In our case, all the embedded points are in $\mathbb{R}^2$, and the embedded points from different clusters are mainly separated by angles; this makes our analysis slightly more involved than [23].

**Analysis for the speeding-up subroutine.** We further analyse the speeding-up subroutine described in Section 4.1. Our analysis is very similar with [27], and the approximation guarantee of our speeding-up subroutine is as follows:

**Theorem 4.5.** *Given a digraph $G = (V, E)$ as input, the speeding-up subroutine computes a subgraph $H = (V, F)$ of $G$ with $O((1/\lambda_2) \cdot n \log n)$ edges. Moreover, with high probability, the computed sparse graph $H$ satisfies that $\theta_k(H) = \Omega(\theta_k(G))$, and $\lambda_2(\mathcal{L}_H) = \Omega(\lambda_2(\mathcal{L}_G))$.*

## 5 Experiments

In this section we present the experimental results of our proposed algorithm `SimpleHerm` on both synthetic and real-world datasets, and compare its performance against the previous state-of-the-art. All our experiments are conducted with an ASUS ZenBook Pro UX501VW with an Intel(R) Core(TM) i7-6700HQ CPU @ 2.60GHz with 12GB of RAM.

We will compare `SimpleHerm` against the `DD-SYM` algorithm [24] and the `Herm-RW` algorithm [9]. Given the adjacency matrix $M \in \mathbb{R}^{n \times n}$ as input, the `DD-SYM` algorithm computes the matrix $A = M^{\mathsf{T}}M + MM^{\mathsf{T}}$, and uses the top $k$ eigenvectors of a random walk matrix $D^{-1}A$ to construct an embedding for $k$-means clustering. The `Herm-RW` algorithm uses the imaginary unit $i$ to represent directed edges and applies the top $\lceil k/2 \rceil$ eigenvectors of a random walk matrix to construct an embedding for $k$-means. Notice that both of the `DD-SYM` and `Herm-RW` algorithms involve the use of multiple eigenvectors, and `DD-SYM` requires computing matrix multiplications, which makes it computationally more expensive than ours.

### 5.1 Results on Synthetic Datasets

We first perform experiments on graphs generated from the Directed Stochastic Block Model (DSBM) which is introduced in [9]. We introduce a path structure into the DSBM, and compare the performance of our algorithm against the others. Specifically, for given parameters $k, n, p, q, \eta$, a graph randomly chosen from the DSBM is constructed as follows: the overall graph consists of $k$ clusters $S_0, \ldots, S_{k-1}$ of the same size, each of which can be initially viewed as a $G(n, p)$ random graph. We connect edges with endpoints in different clusters with probability $q$, and connect edges with endpoints within the same cluster with probability $p$. In addition, for any edge $(u, v)$ where $u \in S_j$ and $v \in S_{j+1}$, we set the edge direction as $u \rightsquigarrow v$ with probability $\eta$, and set the edge direction as $v \rightsquigarrow u$ with probability $1 - \eta$. For all other pairs of clusters which do not lie along the path, we set their edge directions randomly. The directions of edges inside a cluster are assigned randomly.

As graphs generated from the DSBM have a well-defined ground truth clustering, we apply the Adjusted Rand Index (ARI) [14] to measure the performance of different algorithms. We further set $p = q$, since this is one of the hardest regimes for studying the DSBM. In particular, when $p = q$, the edge density plays no role in characterising the structure of clusters, and the edges are *entirely* defined with respect to the edge directions.

We set $n = 1000$, and $k = 4$. We set the value of $p$ to be between $0.5$ and $0.8$, and the value of $\eta$ to be between $0.5$ and $0.7$. As shown in Figure 1, our proposed `SimpleHerm` clearly outperforms the `Herm-RW` and the `DD-SYM` algorithms.

Next, we study the case of $n = 2000$ and $k = 8$, but the structure of clusters presents a more significant path topology. Specifically, we assume that any pair of vertices within each cluster are connected with probability $p \in (0.05, 0.1)$; moreover, all the edges crossing different clusters are along the cuts $(S_j, S_{j+1})$ for some $0 \leqslant j \leqslant k - 2$. By setting $\eta \in (0.65, 1)$, our results are reported in Figure 2. From these results, it is easy to see that, when the underlying graph presents a clear flow structure, our algorithm performs significantly better than both the `Herm-RW` and `DD-SYM` algorithms, for which multiple eigenvectors are needed.

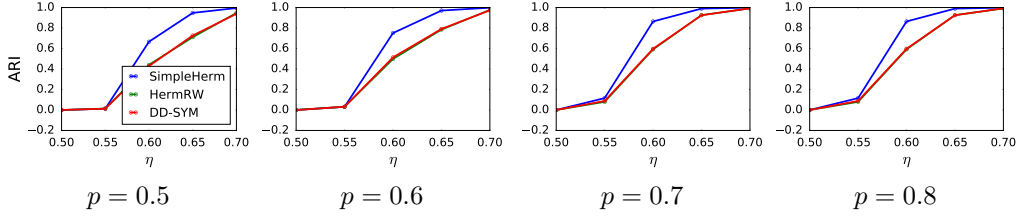

Figure 1: $n = 1000$ and $k = 4$. Average ARIs over 5 runs of different algorithms, with respect to different values of $p$ and $\eta$.

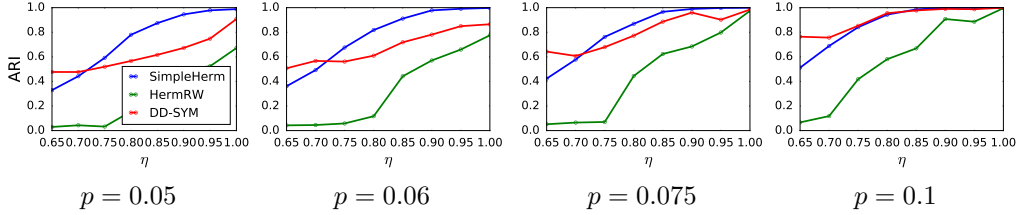

Figure 2: $n = 2000$ and $k = 8$. Average ARIs over 5 runs of different algorithms, with respect to different values of $p$ and $\eta$.

## 5.2 Results on the UN Comtrade Dataset

We compare our proposed algorithm against the previous state-of-the-art on the UN Comtrade Dataset [29]. This dataset consists of the import-export tradeflow data of 97 specific commodities across $N = 246$ countries and territories over the period 1964 – 2018. The total size of the data in zipped files is 99.8GB, where every csv file for a single year contained around 20,000,000 lines.

**Pre-processing.** As the pre-processing step, for any fixed commodity $c$ and any fixed year, we construct a directed graph as follows: the constructed graph has $N = 246$ vertices, which correspond to 246 countries and territories listed in the dataset. For any two vertices $j$ and $\ell$, there is a directed edge from $j$ to $\ell$ if the export of commodity $c$ from country $j$ to $\ell$ is greater than the export from $\ell$ to $j$, and the weight of that edge is set to be the absolute value of the difference in trade, i.e., the net trade value between $\ell$ and $j$. Notice that our construction ensures that all the edge weights are non-negative, and there is at most one directed edge between any pair of vertices.

**Result on the International Oil Trade Industry.** We first study the international trade for mineral fuels, oils, and oil distillation product in the dataset. The primary reason for us to study the international oil trade is due to the fact that crude oil is one of the highest traded commodities worldwide [2], and plays a significant role in geopolitics (e.g., 2003 Iraq War). Many references in international trade and policy making (e.g., [3, 10, 11]) allow us to interpret the results of our proposed algorithm.

Following previous studies on the same dataset from complex networks' perspectives [13, 32], we set $k = 4$. Our algorithm's output around the period of 2006–2009 is visualised in Figure 3. We choose to highlight the results between 2006 and 2009, since 2008 sees the largest post World Ward II oil shock after the economic crisis [18]. As discussed earlier, our algorithm's output is naturally associated with an ordering of the clusters that optimises the value of $\Phi$, and this ordering is reflected in our visualisation as well. Notice that such ordering corresponds to the chain of oil trade, and indicates the clusters of main export countries and import countries for oil trade.

From Figure 3, we see that the output of our algorithm from 2006 to 2008 is pretty stable, and this is in sharp contrast to the drastic change between 2008 and 2009, caused by the economic crisis. Moreover, many European countries move across different clusters from 2008 to 2009. The visualisation results of the other algorithms are less significant than ours.

We further show that this dynamic change of clusters provides a reasonable reflection of international economics. Specifically, we compute the clustering results of our `SimpleHerm` algorithm on the same dataset from 2002 to 2017, and compare it with the output of the `DD-SYM` algorithm. For every two

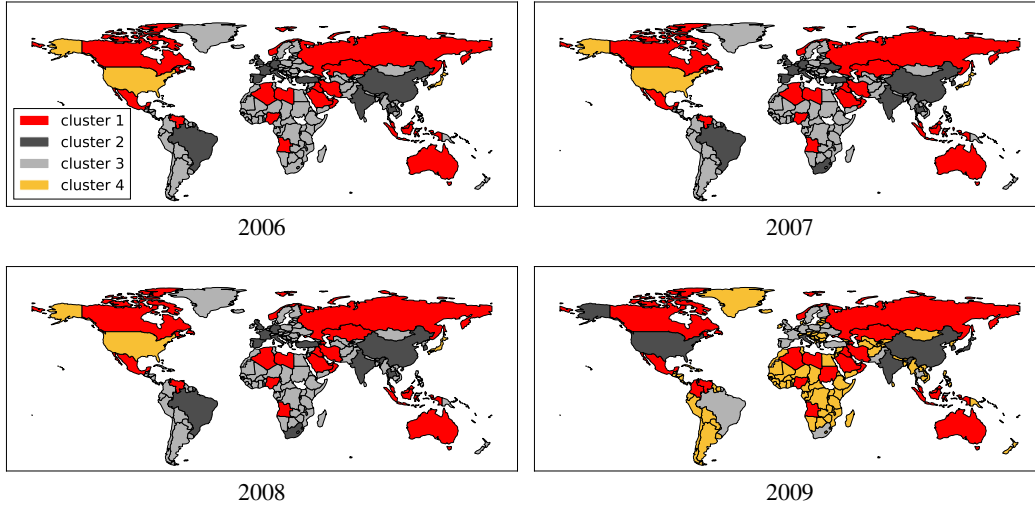

**Figure 3:** The clustering result for international trade from 2006 to 2009, where $k = 4$. Red countries form start of the trade chain, and yellow countries the end of the trade chain. Countries coloured white have no data.

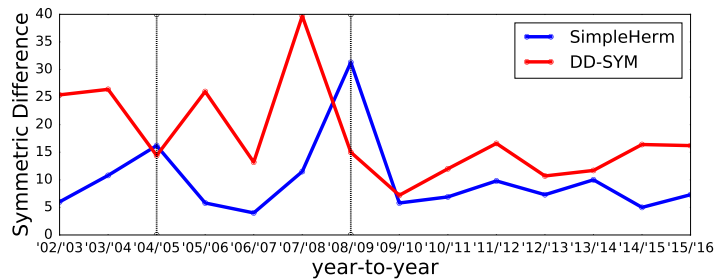

**Figure 4:** Comparison of the symmetric difference of the returned clusters between two consecutive years.

consecutive years, we map every cluster to its "optimal" correspondence (i.e., the one that minimises the symmetric difference between the two). We further compute the total symmetric difference between the clustering results for every two consecutive years, and our results are visualised in Figure 4. As shown in the figure, our algorithm has notable changes in clustering during 2004/2005 and 2008/2009 respectively. The peak around 2004/2005 might be a delayed change as a consequence of the Venezuelan oil strike and the Iraq war of 2003. Both the events led to the decrease in oil barrel production by $5.4$ million barrels per day [16]. The peak around 2008/2009 is of course due to the economic crisis. These peaks correspond to the same periods of cluster instability found in the complex network analysis literature [1, 33], further signifying our result[4]. Compared to our algorithm, the clustering result of the DD-SYM algorithm is less stable over time.

**Result on the International Wood Trade.** We also study the international wood trade network (IWTN). This network looks at the trade of wood and articles of wood. Although the IWTN is less studied than the International Oil Trade Industry in the literature, it is nonetheless the reflection of an important and traditional industry and deserves detailed analysis. Wood trade is dependent on a number of factors, such as the amount of forest a country has left, whether countries are trying to reintroduce forests, and whether countries are deforesting a lot for agriculture (e.g., Amazon rainforest in Brazil) [17].

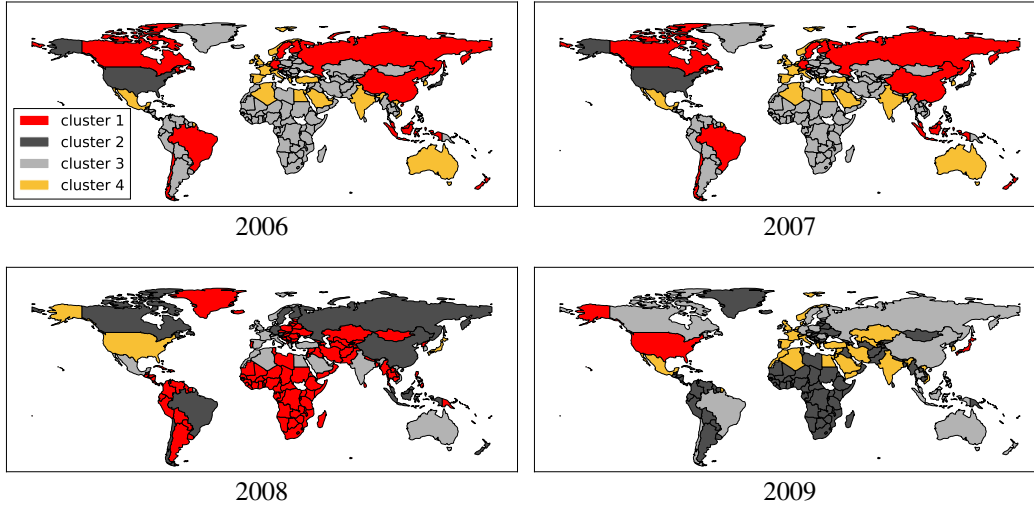

Figure 5: Change in clustering of `SimpleHerm` of the IWTN from 2006 to 2009 with $k = 4$. Clusters are labelled according to their position in the ordering that maximises the flow ratio between the 4 clusters. Red countries form start of the trade chain, and yellow countries the end of the trade chain. Countries coloured in white have no data.

Figure 5 visualises the clusters from 2006 to 2009. As we can see, the structure of clusters are stable in early years, and the first cluster contains countries with large forests such as Canada, Brazil, Russia, Germany, and China. However, there is a significant change of the cluster structure from 2008 to 2009, and countries in Eastern Europe, the Middle East and Central Asia move across different clusters.

### 5.3 Result on the Data Science for COVID-19 Dataset

The Data Science for COVID-19 Dataset (DS4C) [19] contains information about 3519 South Korean citizens infected with COVID-19. Here, digraphs are essential to represent how the virus is transmitted among the individuals, and the clusters with high ratio of out-going edges represent the communities worst hit by the virus. We first identify the largest connected component of the infection graph, which consists of 67 vertices and 66 edges, and run our algorithm on the largest connected component. By setting $k = 4$, our algorithm manages to identify a super-spreader as a single cluster, and the path of infection between groups of people along which most infections lie.

## 6 Broader Impact

The primary focus of our work is efficient clustering algorithms for digraphs, whose clusters are defined with respect to the edge directions between different clusters. We believe that our work could have long-term social impact. For instance, when modelling the transmission of COVID-19 among individuals through a digraph, the cluster (group of people) with the highest ratio of out-going edges represents the most infectious community. This type of information could aid local containment policy. With the development of many tracing Apps for COVID-19 and a significant amount of infection data available in the near future, our studied algorithm could potentially be applied in this context. In addition, as shown by our experimental results on the UN Comtrade Dataset, our work could be employed to analyse many practical data for which most traditional clustering algorithms do not suffice.

## Acknowledgments and Disclosure of Funding

Part of this work was done when Steinar Laenen studied at the University of Edinburgh as a Master student. He Sun is supported by an EPSRC Early Career Fellowship (EP/T00729X/1).

## Footnotes

* steinar9@gmail.com

[2]It is important to notice that, among $2 \cdot \binom{k}{2}$ cuts formed by pairwise different clusters, only $(k - 1)$ cut values contribute to our objective function. If one takes all of the $2 \cdot \binom{k}{2}$ cut values into account, the objective function would involve $2 \cdot \binom{k}{2}$ terms. However, even if most of the $2 \cdot \binom{k}{2}$ terms are much smaller than the ones along the flow, their sum could still be dominant, leaving little information on the structure of clusters. Therefore, we should only take $(k - 1)$ cut values into account when the clusters present a flow structure.

[3]Given any graph $G$ with $n$ vertices and $m$ edges as input, we say an algorithm runs in nearly-linear time if the algorithm's runtime is $O(m \cdot \log^c n)$ for some constant $c$.

[4]We didn't plot the result between 2016 and 2017, since the symmetric difference for the DD-SYM algorithm is 107 and the symmetric difference for the SimpleHerm algorithm is 17. We believe this is an anomaly for DD-SYM, and plotting this result in the same figure would make it difficult to compare other years' results.

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
