[Supplementary Material]

# A   Omitted details from Section 3

In this section we present all the technical detailed omitted from Section 3.

*Proof of Lemma 3.1.*   We prove the statement by analysing the Reyleigh quotient of $\mathcal{L}_G$ with respect to $y$, which is defined by $\frac{y^*\mathcal{L}_G y}{y^* y}$. Since $\|y\| = 1$, it suffices to analyse $y^*\mathcal{L}_G y$. By definition, we have that

$$
\begin{aligned}
y^*\mathcal{L}_G y &= \frac{1}{k}\left(\sum_{j=0}^{k-1}\widehat{\chi_j}\right)^* \mathcal{L}_G \left(\sum_{j=0}^{k-1}\widehat{\chi_j}\right) \\
&= \frac{1}{k}\left(\sum_{j=0}^{k-1}\frac{D^{1/2}\chi_j}{\|D^{1/2}\chi_j\|}\right)^* D^{-1/2}(D-A)D^{-1/2}\left(\sum_{j=0}^{k-1}\frac{D^{1/2}\chi_j}{\|D^{1/2}\chi_j\|}\right) \\
&= \frac{1}{k}\left(\sum_{j=0}^{k-1}\frac{\chi_j}{\|D^{1/2}\chi_j\|}\right)^* (D-A)\left(\sum_{j=0}^{k-1}\frac{\chi_j}{\|D^{1/2}\chi_j\|}\right).
\end{aligned}
\tag{5}
$$

To analyse (5), first of all it is easy to see that

$$
\begin{aligned}
&\frac{1}{k}\left(\sum_{j=0}^{k-1}\frac{\chi_j}{\|D^{1/2}\chi_j\|}\right)^* D \left(\sum_{j=0}^{k-1}\frac{\chi_j}{\|D^{1/2}\chi_j\|}\right) \\
=&\frac{1}{k}\left(\sum_{j=0}^{k-1}\frac{\chi_j}{\|D^{1/2}\chi_j\|}\right)^* D^{1/2}D^{1/2} \left(\sum_{j=0}^{k-1}\frac{\chi_j}{\|D^{1/2}\chi_j\|}\right) \\
=&\frac{1}{k}\left(\sum_{j=0}^{k-1}\frac{D^{1/2}\chi_j}{\|D^{1/2}\chi_j\|}\right)^* \left(\sum_{j=0}^{k-1}\frac{D^{1/2}\chi_j}{\|D^{1/2}\chi_j\|}\right) \\
=&\frac{1}{k}\sum_{j=0}^{k-1}\frac{\chi_j^* D\chi_j}{\|D^{1/2}\chi_j\|^2} \\
=&1,
\end{aligned}
\tag{6}
$$

where the third equality follows by the fact that $\chi_j^*\chi_\ell = 0$ for any $0 \leqslant j \neq \ell \leqslant k-1$. On the other hand, by definition we have that

$$
\begin{aligned}
&\frac{1}{k}\left(\sum_{j=0}^{k-1}\frac{\chi_j}{\|D^{1/2}\chi_j\|}\right)^* A \left(\sum_{j=0}^{k-1}\frac{\chi_j}{\|D^{1/2}\chi_j\|}\right) \\
=&\frac{1}{k}\left(\sum_{j=0}^{k-1}\frac{\chi_j}{\sqrt{\mathrm{vol}\,(S_j)}}\right)^* A \left(\sum_{j=0}^{k-1}\frac{\chi_j}{\sqrt{\mathrm{vol}\,(S_j)}}\right) \\
=&\frac{1}{k}\cdot\sum_{j=0}^{k-1}\sum_{\ell=0}^{k-1}\sum_{\substack{u\rightsquigarrow v \\ u\in S_j, v\in S_\ell}}\left(\frac{\overline{\chi_j}(u)}{\sqrt{\mathrm{vol}\,(S_j)}}\cdot A_{u,v}\cdot\frac{\chi_\ell(v)}{\sqrt{\mathrm{vol}\,(S_\ell)}} + \frac{\overline{\chi_\ell}(v)}{\sqrt{\mathrm{vol}\,(S_\ell)}}\cdot A_{v,u}\cdot\frac{\chi_j(u)}{\sqrt{\mathrm{vol}\,(S_j)}}\right) \\
=&\frac{1}{k}\cdot\sum_{j=0}^{k-1}\sum_{\ell=0}^{k-1}\sum_{\substack{u\rightsquigarrow v \\ u\in S_j, v\in S_\ell}}\frac{w(u,v)}{\sqrt{\mathrm{vol}\,(S_j)}\cdot\sqrt{\mathrm{vol}\,(S_\ell)}}\cdot 2\cdot\mathsf{Re}\left(\left(\omega_{\lceil 2\pi\cdot k\rceil}\right)^{\ell+1-j}\right) \\
=&\frac{1}{k}\cdot\sum_{j=0}^{k-1}\sum_{\ell=0}^{k-1}\frac{w\,(S_j, S_\ell)}{\sqrt{\mathrm{vol}\,(S_j)}\cdot\sqrt{\mathrm{vol}\,(S_\ell)}}\cdot 2\cdot\cos\left(\frac{2\pi\cdot(\ell+1-j)}{\lceil 2\pi\cdot k\rceil}\right),
\end{aligned}
\tag{7}
$$

where $\mathrm{Re}(\cdot)$ stands for the real part of a complex number. Combining (5), (6) with (7), we have that
$y^* \mathcal{L}_G y$

$$= 1 - \frac{1}{k} \cdot \sum_{j=0}^{k-1} \sum_{\ell=0}^{k-1} \frac{w\left(S_j, S_\ell\right)}{\sqrt{\mathrm{vol}\left(S_j\right)} \cdot \sqrt{\mathrm{vol}\left(S_\ell\right)}} \cdot 2 \cdot \cos\left(\frac{2\pi \cdot (\ell + 1 - j)}{\lceil 2\pi \cdot k \rceil}\right)$$

$$\leqslant 1 - \frac{1}{k} \cdot \sum_{j=0}^{k-1} \sum_{\ell=0}^{k-1} \frac{w\left(S_j, S_\ell\right)}{\sqrt{\mathrm{vol}(S_j)} \sqrt{\mathrm{vol}(S_\ell)}} \cdot \left(2 - \left(\frac{2\pi \cdot (\ell + 1 - j)}{\lceil 2\pi \cdot k \rceil}\right)^2\right)$$

$$\leqslant 1 - \frac{1}{k} \cdot \sum_{j=0}^{k-1} \sum_{\ell=0}^{k-1} \frac{2 \cdot w\left(S_j, S_\ell\right)}{\sqrt{\mathrm{vol}(S_j)} \sqrt{\mathrm{vol}(S_\ell)}} + \frac{1}{k} \cdot \sum_{j=0}^{k-1} \sum_{\ell=0}^{k-1} \frac{w\left(S_j, S_\ell\right)}{\sqrt{\mathrm{vol}(S_j)} \sqrt{\mathrm{vol}(S_\ell)}} \left(\frac{\ell + 1 - j}{k}\right)^2$$

$$\leqslant 1 - \frac{1}{k} \cdot \sum_{j=0}^{k-1} \sum_{\ell=0}^{k-1} \frac{2 \cdot w\left(S_j, S_\ell\right)}{\sqrt{\mathrm{vol}(S_j)} \sqrt{\mathrm{vol}(S_\ell)}} + \frac{1}{k} \cdot \sum_{j=0}^{k-1} \sum_{\substack{0 \leqslant \ell \leqslant k-1 \\ \ell \neq j-1}} \frac{2 \cdot w\left(S_j, S_\ell\right)}{\sqrt{\mathrm{vol}(S_j)} \sqrt{\mathrm{vol}(S_\ell)}} \left(\frac{\ell + 1 - j}{k}\right)^2$$

$$= 1 - \frac{1}{k} \cdot \sum_{j=0}^{k-1} \sum_{\substack{0 \leqslant \ell \leqslant k-1 \\ \ell \neq j-1}} \frac{2 \cdot w(S_j, S_\ell)}{\sqrt{\mathrm{vol}(S_j)} \sqrt{\mathrm{vol}(S_\ell)}} \left(1 - \left(\frac{\ell + 1 - j}{k}\right)^2\right) - \frac{1}{k} \cdot \sum_{j=1}^{k-1} \frac{2 \cdot w(S_j, S_{j-1})}{\sqrt{\mathrm{vol}(S_j)} \sqrt{\mathrm{vol}(S_{j-1})}}$$

$$\leqslant 1 - \frac{1}{k} \cdot \sum_{j=1}^{k-1} \frac{2 \cdot w(S_j, S_{j-1})}{\sqrt{\mathrm{vol}(S_j)} \sqrt{\mathrm{vol}(S_{j-1})}}$$

$$= 1 - \frac{2}{k} \cdot \sum_{j=1}^{k-1} \frac{w(S_j, S_{j-1})}{\sqrt{\mathrm{vol}(S_j)} \sqrt{\mathrm{vol}(S_{j-1})}}$$

$$\leqslant 1 - \frac{4}{k} \cdot \sum_{j=1}^{k-1} \frac{w(S_j, S_{j-1})}{\mathrm{vol}(S_j) + \mathrm{vol}(S_{j-1})}$$

$$= 1 - \frac{4}{k} \cdot \theta_k(G),$$

where the first inequality follows by the fact that $\cos x \geqslant 1 - x^2/2$ and the last inequality follows by the inequality $2ab \leqslant a^2 + b^2$ for any $a, b \in \mathbb{R}$. Therefore, we have that

$$\frac{y^* \mathcal{L}_G y}{y^* y} \leqslant 1 - \frac{4}{k} \cdot \theta_k(G).$$

By the Rayleigh characterisation of eigenvalues we know that

$$\lambda_1(\mathcal{L}_G) = \min_{x \in \mathbb{C}^n \setminus \{0\}} \frac{x^* \mathcal{L}_G x}{x^* x} \leqslant 1 - \frac{4}{k} \cdot \theta_k(G),$$

which proves the first statement of the lemma.

Now we prove the second statement. Let $G$ be a digraph, and $S_0, \ldots, S_{k-1}$ be the $k$ clusters maximising $\Phi_G(S_0, \ldots, S_{k-1})$, i.e., $\Phi_G(S_0, \ldots, S_{k-1}) = \theta_k(G)$. Since adding edges that are not along the path only decreases the value of $\Phi_G$, we assume without loss of generality that all the edges are along the path. For the base case of $k = 2$, we have that

$$\Phi_G(S_0, S_1) = \frac{w(S_0, S_1)}{\mathrm{vol}(S_0) + \mathrm{vol}(S_1)} = \frac{1}{2} = \frac{k}{4}.$$

Next, we will prove that $\theta_k(G) < k/4$ for any $k \geqslant 3$. We set $y_j \triangleq w(S_j, S_{j-1})$ for any $1 \leqslant j \leqslant k-1$, and have that

$$\Phi_G(S_0, \ldots, S_{k-1}) = \sum_{j=1}^{k-1} \frac{w(S_j, S_{j-1})}{\mathrm{vol}(S_j) + \mathrm{vol}(S_{j-1})}$$

$$= \frac{y_1}{2y_1 + y_2} + \sum_{j=2}^{k-2} \frac{y_j}{y_{j-1} + 2y_j + y_{j+1}} + \frac{y_{k-1}}{y_{k-2} + 2y_{k-1}}.$$

By introducing $y_0 = 0$ and assuming that all the indices of $\{y_j\}_j$ are modulo b $k$, we can write $\Phi_G(S_0, \ldots, S_{k-1})$ as

$$\Phi_G(S_0, \ldots, S_{k-1}) = \sum_{j=0}^{k-1} \frac{y_j}{y_{j-1} + 2y_j + y_{j+1}}.$$

Next we compute $\partial \Phi_G / \partial y_j$, and have that

$$\frac{\partial \Phi_G}{\partial y_j} = \frac{\partial \Phi_G}{\partial y_j} \sum_{j=0}^{k-1} \frac{y_j}{y_{j-1} + 2y_j + y_{j+1}}$$

$$= \frac{\partial \Phi_G}{\partial y_j} \left( \frac{y_{j-1}}{y_{j-2} + 2y_{j-1} + y_j} + \frac{y_j}{y_{j-1} + 2y_j + y_{j+1}} + \frac{y_{j+1}}{y_j + 2y_{j+1} + y_{j+2}} \right)$$

$$= -\frac{y_{j-1}}{(y_{j-2} + 2y_{j-1} + y_j)^2} + \frac{y_{j-1} + y_{j+1}}{(y_{j-1} + 2y_j + y_{j+1})^2} - \frac{y_{j+1}}{(y_j + 2y_{j+1} + y_{j+2})^2}.$$

Notice that, when all the $y_j$ ($0 \leqslant j \leqslant k-1$) equal to the same non-zero value, it holds that $\partial \Phi_G / \partial y_j = 0$ for any $j$, and $\theta_G(S_0, \ldots, S_{k-1}) = k/4$. Moreover, it's easy to verify that $k/4$ is an upper bound of $\theta_G$. Since we effectively assume that $y_0 = 0$, which cannot be always equal to all of the $y_1, \ldots, y_{k-1}$, we have that $\theta_G(S_0, \ldots, S_{k-1}) < k/4$. $\qquad \square$

*Proof of Theorem 3.2.* We first prove the first statement. We write $y$ as a linear combination of the eigenvectors of $\mathcal{L}_G$ by

$$y = \alpha_1 f_1 + \cdots + \alpha_n f_n$$

for some $\alpha_i \in \mathbb{C}$ and $f_i \in \mathbb{C}^n$, and define $\widetilde{f_1}$ by $\widetilde{f_1} \triangleq \alpha_1 f_1$. By the definition of the Rayleigh quotient for Hermitian matrices we have that

$$\frac{y^* \mathcal{L}_G y}{\|y\|} = (\alpha_1 f_1 + \cdots + \alpha_n f_n)^* \mathcal{L}_G (\alpha_1 f_1 + \cdots + \alpha_n f_n)$$

$$= \|\alpha_1\|^2 \lambda_1(\mathcal{L}_G) + \cdots + \|\alpha_n\|^2 \lambda_n(\mathcal{L}_G)$$

$$\geqslant \|\alpha_1\|^2 \lambda_1(\mathcal{L}_G) + (\|\alpha_2\|^2 + \cdots + \|\alpha_n\|^2) \lambda_2(\mathcal{L}_G)$$

$$\geqslant (1 - \|\alpha_1\|^2) \lambda_2(\mathcal{L}_G),$$

where the first inequality holds by the fact that $\lambda_1(\mathcal{L}_G) \leqslant \ldots \leqslant \lambda_n(\mathcal{L}_G)$ and the second inequality holds by the fact that $\|\alpha_2\|^2 + \cdots + \|\alpha_n\|^2 = 1 - \|\alpha_1\|^2$. We can see that

$$\left\| y - \widetilde{f_1} \right\|^2 = \|\alpha_2\|^2 + \cdots + \|\alpha_n\|^2 = 1 - \|\alpha_1\|^2 \leqslant \frac{1}{\lambda_2} \cdot \frac{y^* \mathcal{L}_G y}{\|y\|} \leqslant \frac{1}{\gamma_k(G)}.$$

Setting $\alpha = \alpha_1$ proves the first statement.

Next we prove the second statement. By the relationship between $f_1$ and $\widetilde{f_1}$, we write

$$f_1 = \beta_1 \widetilde{f_1},$$

where $\beta_1 \triangleq 1/\alpha_1$ is the multiplicative inverse of $\alpha_1$. Then, we define $\widetilde{y}$ as

$$\widetilde{y} = \beta_1 y = \beta_1 (\alpha_1 f_1 + \cdots + \alpha_n f_n) = f_1 + \beta_1 (\alpha_2 f_2 + \cdots + \alpha_n f_n),$$

and this implies that

$$\|f_1 - \widetilde{y}\|^2 = \|\beta_1 (\alpha_2 f_2 + \cdots + \alpha_n f_n)\|^2 = \overline{\beta_1} \cdot \left( \sum_{j=2}^n \|\alpha_j\|^2 \right) \cdot \beta_1 = \frac{1}{\|\alpha_1\|^2} \left( 1 - \|\alpha_1\|^2 \right)$$

$$\leqslant \frac{1}{\|\alpha_1\|^2 \cdot \gamma_k(G)}. \tag{8}$$

Since $1 - \|\alpha_1\|^2 \leqslant 1/\gamma_k(G)$ implies that

$$\|\alpha_1\|^2 \geqslant \frac{\gamma_k(G) - 1}{\gamma_k(G)},$$

we can rewrite (8) as

$$\|f_1 - \widetilde{y}\|^2 \leqslant \frac{1}{\gamma_k(G) - 1},$$

and therefore setting $\beta = \beta_1$ proves the second statement. $\qquad \square$

# B  Omitted details from Section 4

In this section we present all the technical detailed omitted from Section 4.

*Proof of Lemma 4.1.* By definition, we have that

$$
\sum_{j=0}^{k-1}\sum_{u\in S_j} d_u\cdot\left\|F(u)-p^{(j)}\right\|^2 = \sum_{j=0}^{k-1}\sum_{u\in S_j} d_u\cdot\left\|\frac{1}{\sqrt{d_u}}\cdot f_1(u)-\frac{\beta}{\sqrt{k}}\cdot\frac{(\omega_{\lceil 2\pi\cdot k\rceil})^j}{\sqrt{\mathrm{vol}(S_j)}}\right\|^2
$$

$$
=\sum_{j=0}^{k-1}\sum_{u\in S_j}\left\|f_1(u)-\sqrt{\frac{d_u}{k\cdot\mathrm{vol}(S_j)}}\cdot\beta\cdot(\omega_{\lceil 2\pi\cdot k\rceil})^j\right\|^2
$$

$$
=\sum_{j=0}^{k-1}\sum_{u\in S_j}\left\|f_1(u)-\widetilde{y}(u)\right\|^2
$$

$$
=\|f_1-\widetilde{y}\|^2
$$

$$
\leqslant\frac{1}{\gamma_k(G)-1},
$$

where the last inequality follows by Theorem 3.2. $\qquad\square$

*Proof of Lemma 4.2.* The proof is by direct calculation on $\left\|p^{(j)}\right\|^2$. $\qquad\square$

*Proof of Lemma 4.3.* By definition of $p^{(j)}$ and $p^{(\ell)}$, we have that

$$
\left\|p^{(j)}-p^{(\ell)}\right\|^2
$$

$$
=\left\|p^{(j)}\right\|^2+\left\|p^{(\ell)}\right\|^2-2\cdot\mathrm{Re}\left(p^{(j)}\cdot\overline{p^{(\ell)}}\right)
$$

$$
=\frac{\|\beta\|^2}{k\cdot\mathrm{vol}(S_j)}+\frac{\|\beta\|^2}{k\cdot\mathrm{vol}(S_\ell)}-2\cdot\mathrm{Re}\left(\frac{\beta\cdot(\omega_{\lceil 2\pi\cdot k\rceil})^j}{\sqrt{k}\cdot\sqrt{\mathrm{vol}(S_j)}}\cdot\frac{\overline{\beta}\cdot(\omega_{\lceil 2\pi\cdot k\rceil})^{-\ell}}{\sqrt{k}\cdot\sqrt{\mathrm{vol}(S_\ell)}}\right)
$$

$$
=\frac{\|\beta\|^2}{k\cdot\mathrm{vol}(S_j)}+\frac{\|\beta\|^2}{k\cdot\mathrm{vol}(S_\ell)}-2\cdot\frac{\|\beta\|^2}{k\cdot\sqrt{\mathrm{vol}(S_j)\cdot\mathrm{vol}(S_\ell)}}\cdot\cos\left(\frac{2\pi\cdot(j-\ell)}{\lceil 2\pi\cdot k\rceil}\right). \qquad (9)
$$

For the case of calculation and the fact that $\cos(x)=\cos(-x)$ for any $x\in\mathbb{R}$, we denote

$$
\eta\triangleq\frac{2\pi\cdot|j-\ell|}{\lceil 2\pi\cdot k\rceil},
$$

and rewrite (9) as

$$
\left\|p^{(j)}-p^{(\ell)}\right\|^2
$$

$$
=\frac{\|\beta\|^2}{k\cdot\mathrm{vol}(S_j)}+\frac{\|\beta\|^2}{k\cdot\mathrm{vol}(S_\ell)}-2\cdot\frac{\|\beta\|^2}{k\cdot\sqrt{\mathrm{vol}(S_j)\cdot\mathrm{vol}(S_\ell)}}\cdot\cos\eta
$$

$$
=\frac{\|\beta\|^2}{k\cdot\max\{\mathrm{vol}(S_j),\mathrm{vol}(S_\ell)\}}+\frac{\|\beta\|^2\cdot(\sin^2\eta+\cos^2\eta)}{k\cdot\min\{\mathrm{vol}(S_j),\mathrm{vol}(S_\ell)\}}-\frac{2\cos\eta\cdot\|\beta\|^2}{k\cdot\sqrt{\mathrm{vol}(S_j)\cdot\mathrm{vol}(S_\ell)}}
$$

$$
=\left(\frac{\|\beta\|}{\sqrt{k\cdot\max\{\mathrm{vol}(S_j),\mathrm{vol}(S_\ell)\}}}-\frac{\cos\eta\cdot\|\beta\|}{\sqrt{k\cdot\min\{\mathrm{vol}(S_j),\mathrm{vol}(S_\ell)\}}}\right)^2+\frac{\|\beta\|^2\cdot\sin^2\eta}{k\cdot\min\{\mathrm{vol}(S_j),\mathrm{vol}(S_\ell)\}}
$$

$$
\geqslant\frac{\|\beta\|^2\cdot\sin^2\eta}{k\cdot\min\{\mathrm{vol}(S_j),\mathrm{vol}(S_\ell)\}}
$$

$$
\geqslant\frac{\|\beta\|^2}{k\cdot\min\{\mathrm{vol}(S_j),\mathrm{vol}(S_\ell)\}}\cdot\left(\frac{2\pi\cdot|j-\ell|}{\lceil 2\pi\cdot k\rceil}\cdot\frac{2}{\pi}\right)^2
$$

$$
\geqslant\frac{\|\beta\|^2}{3k^3\cdot\min\{\mathrm{vol}(S_j),\mathrm{vol}(S_\ell)\}},
$$

where the second inequality holds by the fact that $\sin x \geqslant (2/\pi) \cdot x$ holds for any $x \in [0, \pi/2]$. This finishes the proof of the lemma. $\qquad\qquad\qquad\qquad\qquad\qquad\qquad\qquad\qquad\qquad\qquad\qquad\qquad$ $\square$

The following lemma will be used to prove Theorem 4.4. We remark that the following proof closely follows the similar one from [23], however some constants need to be adjusted for our propose. We include the proof here for completeness.

**Lemma B.1.** *Let* $A_0, \ldots, A_{k-1}$ *be a partition of* $V$. *Assume that, for every permutation* $\sigma : \{0, \ldots, k-1\} \rightarrow \{0, \ldots, k-1\}$, *there exists some* $j'$ *such that* $\mathrm{vol}\left(A_{j'} \triangle S_{\sigma(j')}\right) \geqslant \varepsilon \mathrm{vol}\left(S_{\sigma(j')}\right)$ *for some* $48 \cdot k^3 \cdot (1 + \mathsf{APT}) \big/ (\gamma_k(G) - 1) \leqslant \varepsilon \leqslant 1/2$, *then* $\mathsf{COST}(A_0, \ldots, A_{k-1}) \geqslant 2\mathsf{APT} \big/ (\gamma_k(G) - 1)$.

*Proof.* We first consider the case where there exists a permutation $\sigma : \{0, \ldots, k-1\} \rightarrow \{0, \ldots, k-1\}$ such that, for any $0 \leqslant j \leqslant k - 1$,

$$\mathrm{vol}\left(A_j \cap S_{\sigma(j)}\right) > \frac{1}{2}\mathrm{vol}\left(S_{\sigma(j)}\right). \tag{10}$$

This assumption essentially says that $A_0, \ldots, A_{k-1}$ is a non-trivial approximation of the optimal clustering $S_0, \ldots, S_{k-1}$ according to some permutation $\sigma$. Later we will show the statement of the lemma trivially holds if no permutations satisfy (10).

Based on this assumption, there is $0 \leqslant j' \leqslant k - 1$ such that $\mathrm{vol}\left(A_{j'} \triangle S_{\sigma(j')}\right) \geqslant 2\varepsilon \mathrm{vol}\left(S_{\sigma(j')}\right)$ for some $48 \cdot k^3 \cdot \mathsf{APT} \big/ (\gamma_k(G) - 1) \leqslant \varepsilon \leqslant 1/2$. Since

$$\begin{aligned}\mathrm{vol}\left(A_{j'} \triangle S_{\sigma(j')}\right) &= \mathrm{vol}\left(A_{j'} \setminus S_{\sigma(j')}\right) + \mathrm{vol}\left(S_{\sigma(j')} \setminus A_{j'}\right) \\ &= \sum_{j \neq j'} \mathrm{vol}\left(A_{j'} \cap S_{\sigma(j)}\right) + \sum_{j \neq j'} \mathrm{vol}\left(S_{\sigma(j')} \cap A_j\right),\end{aligned}$$

one of the following two cases must hold:

1. A large portion of $A_{j'}$ belongs to clusters different from $S_{\sigma(j')}$, i.e., there exist $\varepsilon_0, \ldots, \varepsilon_{k-1} \geqslant 0$ such that $\varepsilon_{j'} = 0$, $\sum_{j=0}^{k-1} \varepsilon_j \geqslant \varepsilon$, and $\mathrm{vol}\left(A_{j'} \cap S_{\sigma(j)}\right) \geqslant \varepsilon_j \mathrm{vol}\left(S_{\sigma(j')}\right)$ for any $0 \leqslant j \leqslant k - 1$.

2. $A_{j'}$ is missing a large portion of $S_{\sigma(j')}$, which must have been assigned to other clusters. Therefore, we can define $\varepsilon_0, \ldots, \varepsilon_{k-1} \geqslant 0$ such that $\varepsilon_{j'} = 0$, $\sum_{j=0}^{k-1} \varepsilon_j \geqslant \varepsilon$, and $\mathrm{vol}\left(A_j \cap S_{\sigma(j')}\right) \geqslant \varepsilon_j \mathrm{vol}\left(S_{\sigma(j')}\right)$ for any $0 \leqslant j \leqslant k - 1$.

In both cases, we can define sets $B_0, \ldots, B_{k-1}$ and $D_0, \ldots, D_{k-1}$ such that $B_j$ and $D_j$ belong to the same cluster of the returned clustering but to two different optimal clusters $S_{j_1}$ and $S_{j_2}$. More precisely, in the first case, for any $0 \leqslant j \leqslant k - 1$, we define $B_j = A_{j'} \cap S_{\sigma(j)}$. We define $D_0, \ldots, D_{k-1}$ as an arbitrarily partition of $A_{j'} \cap S_{\sigma(j')}$ with the constraint that $\mathrm{vol}(D_j) \geqslant \varepsilon_j \mathrm{vol}(S_{\sigma(j')})$. This is possible since by (10)

$$\mathrm{vol}\left(A_{j'} \cap S_{\sigma(j')}\right) \geqslant \frac{1}{2}\mathrm{vol}\left(S_{\sigma(j')}\right) \geqslant \varepsilon \mathrm{vol}\left(S_{\sigma(j')}\right).$$

In the second case, instead, for any $0 \leqslant j \leqslant k - 1$, we define $B_j = A_j \cap S_{\sigma(j')}$ and $D_j = A_j \cap S_{\sigma(j)}$. Note that it also holds by (10) that $\mathrm{vol}(D_j) \geqslant \varepsilon_j \mathrm{vol}(S_{\sigma(j)})$. We can then combine the two cases together (albeit using different definitions for the sets) and assume that there exist $\varepsilon_0, \ldots, \varepsilon_{k-1} \geqslant 0$ such that $\varepsilon_{j'} = 0$, $\sum_{j=0}^{k-1} \varepsilon_j \geqslant \varepsilon$, and such that we can find collections of pairwise disjoint sets $\{B_0, \ldots, B_{k-1}\}$ and $\{D_0, \ldots, D_{k-1}\}$ with the following properties: for any $j$ there exist indices $\bar{j}$ and $j_1 \neq j_2$ such that

1. $B_j, D_j \subseteq A_{\bar{j}}$

2. $D_j \subseteq S_{j_1}, B_j \subseteq S_{j_2}$

3. $\mathrm{vol}(B_j) \geqslant \varepsilon_j \min\{\mathrm{vol}\left(S_{j_1}\right), \mathrm{vol}\left(S_{j_2}\right)\}$

4. $\text{vol}(D_j) \geqslant \varepsilon_j \min\{\text{vol}(S_{j_1}), \text{vol}(S_{j_2})\}$

For any $j$, we define $c_j$ as the centre of the corresponding cluster $A_{\bar{j}}$ to which both $B_j$ and $D_j$ are subset of. We can also assume without loss of generality that $\left\|c_j - p^{(j_1)}\right\| \geqslant \left\|c_j - p^{(j_2)}\right\|$ which implies

$$\left\|p^{(j_1)} - c_j\right\| \geqslant \frac{1}{2} \cdot \left\|p^{(j_1)} - p^{(j_2)}\right\|.$$

As a consequence, points in $B_j$ are far away from $c_j$. Notice that if instead $\left\|c_j - p^{(j_1)}\right\| < \left\|c_j - p^{(j_2)}\right\|$, we would just need to reverse the role of $B_j$ and $D_j$ without changing the proof. We now bound $\mathsf{COST}(A_0, \ldots, A_{k-1})$ by looking only at the contribution of the points in the $B_j$'s. Therefore, we have that

$$\mathsf{COST}(A_0, \ldots, A_{k-1}) = \sum_{j=0}^{k-1} \sum_{u \in A_j} d_u \|F(u) - c_j\|^2 \geqslant \sum_{j=0}^{k-1} \sum_{u \in B_j} d_u \|F(u) - c_j\|^2.$$

By applying the inequality $a^2 + b^2 \geqslant (a - b)^2/2$, we have that

$$\mathsf{COST}(A_0, \ldots, A_{k-1}) \geqslant \sum_{j=0}^{k-1} \sum_{u \in B_j} d_u \left( \frac{\left\|p^{(j_1)} - c_j\right\|^2}{2} - \left\|F(u) - p^{(j_1)}\right\|^2 \right)$$

$$\geqslant \sum_{j=0}^{k-1} \sum_{u \in B_j} d_u \frac{\left\|p^{(j_1)} - c_j\right\|^2}{2} - \sum_{j=0}^{k-1} \sum_{u \in B_j} d_u \left\|F(u) - p^{(j_1)}\right\|^2$$

$$\geqslant \sum_{j=0}^{k-1} \sum_{u \in B_j} d_u \frac{\left\|p^{(j_1)} - c_j\right\|^2}{2} - \frac{1}{\gamma_k(G) - 1}$$

$$\geqslant \sum_{j=0}^{k-1} \sum_{u \in B_j} d_u \frac{\left\|p^{(j_1)} - p^{(j_2)}\right\|^2}{8} - \frac{1}{\gamma_k(G) - 1}$$

$$\geqslant \sum_{j=0}^{k-1} \frac{\|\beta\|^2 \cdot \text{vol}(B_j)}{24k^3 \cdot \min\{\text{vol}(S_{j_1}), \text{vol}(S_{j_2})\}} - \frac{1}{\gamma_k(G) - 1}$$

$$\geqslant \sum_{j=0}^{k-1} \frac{\|\beta\|^2 \cdot \varepsilon_j \min\{\text{vol}(S_{j_1}), \text{vol}(S_{j_2})\}}{24k^3 \cdot \min\{\text{vol}(S_{j_1}), \text{vol}(S_{j_2})\}} - \frac{1}{\gamma_k(G) - 1}$$

$$\geqslant \sum_{j=0}^{k-1} \frac{\varepsilon_j \cdot \|\beta\|^2}{24k^3} - \frac{1}{\gamma_k(G) - 1}$$

$$\geqslant \frac{\varepsilon}{24k^3} - \frac{1}{\gamma_k(G) - 1}$$

$$\geqslant \frac{1}{24k^3} \cdot \frac{48k^3 \cdot (1 + \mathsf{APT})}{\gamma_k(G) - 1} - \frac{1}{\gamma_k(G) - 1}$$

$$\geqslant \frac{2\mathsf{APT}}{\gamma_k(G) - 1}.$$

It remains to show that removing assumption (10) implies the Lemma as well. Notice that if (10) is not satisfied, for all permutations $\sigma$ there exists $0 \leqslant \ell^\star \leqslant k - 1$ such that $\text{vol}\left(A_{\ell^\star} \cap S_{\sigma(\ell^\star)}\right) \leqslant \frac{1}{2}\text{vol}\left(S_{\sigma(\ell^\star)}\right)$. We can also assume the following stronger condition:

$$\text{vol}(A_{\ell^\star} \cap S_j) \leqslant \frac{1}{2}\text{vol}(S_j) \qquad \text{for any } 0 \leqslant j \leqslant k - 1. \tag{11}$$

Indeed, if there would exist a unique $j \neq \sigma(\ell^\star)$ such that $\text{vol}(A_{\ell^\star} \cap S_j) > \frac{1}{2}\text{vol}(S_j)$, then it would just mean that $\sigma$ is the "wrong" permutation and we should consider only permutations $\sigma' \neq \sigma$ such that $\sigma'(\ell^\star) = j$. If instead there would exist $j_1 \neq j_2$ such that $\text{vol}(A_{\ell^\star} \cap S_{j_1}) > \frac{1}{2}\text{vol}(S_{j_1})$ and

$\text{vol}\,(A_{\ell^\star} \cap S_{j_2}) > \frac{1}{2}\text{vol}\,(S_{j_2})$, then it is easy to see that the Lemma would hold, since in this case $A_{\ell^\star}$ would contain large portions of two different optimal clusters, and, as clear from the previous part of the proof, this would imply a high $k$-means cost.

Therefore, we just need to show that the statement of the Lemma holds when (11) is satisfied. For this purpose we define sets $C_0, \ldots, C_{k-1}$ which are subsets of vertices in $S_0, \ldots, S_{k-1}$ that are close in the spectral embedding to $p^{(0)}, \ldots, p^{(k-1)}$. Formally, for any $0 \leqslant j \leqslant k-1$,

$$ C_j = \left\{ u \in S_j : \; \|F(u) - p^{(j)}\|^2 \leqslant \frac{100}{\text{vol}(S_j)} \cdot (\gamma_k(G) - 1) \right\}. $$

Notice that by Lemma 4.1 $\text{vol}(C_j) \geqslant \frac{99}{100}\text{vol}(S_j)$. By assumption (11), roughly half of the volume of all the $C_j$'s must be contained in at most $k-1$ sets (all the $A_j$'s different from $A_{\ell^\star}$). We prove this implies that the $k$-means cost is high, from which the Lemma follows.

Let $c_0, \ldots, c_{k-1}$ be the centres of $A_0, \ldots, A_{k-1}$. We are trying to assign a large portion of each of the $k$ optimal clusters to only $k-1$ centres (namely all the centres different from $c_{\ell^\star}$). Moreover, any centre $c_j \neq c_{\ell^\star}$ can either be close to $p^{(\ell^\star)}$ or to another optimal centre $p^{(j')}$, but not to both. As a result, there will be at least one $C_j$ whose points are assigned to a centre which is at least $\Omega(1/\text{vol}(S_j))$ far from $p^{(j)}$ (in squared Euclidean distance). Therefore, by the definition of $C_j$ and the fact that $\text{vol}(C_j) \geqslant \frac{99}{100}\text{vol}(S_j)$, the $k$-means cost is at least $\Omega\left(\frac{1}{\text{vol}(S_j)} \cdot \text{vol}(C_j)\right) = \Omega(1)$. This concludes the proof. $\qquad\square$

*Proof of Theorem 4.4.* Assume for contradiction that, for any permutation $\sigma : \{0, \ldots, k-1\} \to \{0, \ldots, k-1\}$, there is an index $j \in \{0, \ldots, k-1\}$ such that $\text{vol}\left(A_j \triangle S_{\sigma(j)}\right) \geqslant \varepsilon \text{vol}\left(S_{\sigma(j)}\right)$. Then, by Lemma B.1 we have that $\mathsf{COST}(A_0, \ldots, A_{k-1}) \geqslant 2\mathsf{APT}/\left(\gamma_k(G) - 1\right)$, which contradicts the fact that $\mathsf{COST}(A_0, \ldots, A_{k-1}) \leqslant \mathsf{APT}/\left(\gamma_k(G) - 1\right)$. $\qquad\square$

Now we prove Theorem 4.5. The following two technical lemmas will be used in our proof.

**Lemma B.2** (Bernstein's Inequality, [8]). *Let $X_1, \ldots X_n$ be independent random variables such that $|X_i| \leqslant M$ for any $i \in \{1, \ldots, n\}$. Let $X = \sum_{i=1}^n X_i$ and let $R = \sum_{i=1}^n \mathbb{E}[X_i^2]$. Then, it holds that*

$$ \mathbb{P}\left[|X - \mathbb{E}[X]| \geqslant t\right] \leqslant 2 \cdot \exp\left(-\frac{t^2}{2(R + Mt/3)}\right). $$

**Lemma B.3** (Matrix Chernoff Bound, [28]). *Consider a finite sequence $\{X_i\}$ of independent, random, PSD matrices of dimension $d$ that satisfy $\|X_i\| \leqslant R$. Let $\mu_{\min} \triangleq \lambda_{\min}\left(\mathbb{E}\left[\sum_i X_i\right]\right)$ and $\mu_{\max} \triangleq \lambda_{\max}\left(\mathbb{E}\left[\sum_i X_i\right]\right)$. Then it holds that*

$$ \mathbb{P}\left[\lambda_{\min}\left(\sum_i X_i\right) \leqslant (1-\delta)\mu_{\min}\right] \leqslant d \cdot \left(\frac{\mathrm{e}^{-\delta}}{(1-\delta)^{1-\delta}}\right)^{\mu_{\min}/R} \quad \text{for } \delta \in [0, 1], \text{ and} $$

$$ \mathbb{P}\left[\lambda_{\max}\left(\sum_i X_i\right) \geqslant (1+\delta)\mu_{\max}\right] \leqslant d \cdot \left(\frac{\mathrm{e}^{\delta}}{(1+\delta)^{1+\delta}}\right)^{\mu_{\max}/R} \quad \text{for } \delta \geqslant 0. $$

*Proof of Theorem 4.5.* We first analyse the size of $F$. Since

$$ \sum_{u \in V} \sum_{e=(u,v)} w(u,v) \cdot \frac{\alpha \log n}{d_u^{\text{out}} \cdot \lambda_2} = O\left(\frac{n \log n}{\lambda_2}\right), $$

and

$$ \sum_{v \in V} \sum_{e=(u,v)} w(u,v) \cdot \frac{\alpha \log n}{d_v^{\text{in}} \cdot \lambda_2} = O\left(\frac{n \log n}{\lambda_2}\right), $$

it holds by Markov inequality that the number of edges $e = (u,v)$ with $w(u,v) \cdot \frac{\alpha \log n}{d_u^{\text{out}} \cdot \lambda_2} \geqslant 1$ and $w(u,v) \cdot \frac{\alpha \log n}{d_v^{\text{in}} \cdot \lambda_2} \geqslant 1$ is $O\left(\frac{n \log n}{\lambda_2}\right)$. Without loss of generality, we assume that these edges are in $F$, and in the remaining part of the proof we assume it holds for any edge $e = (u,v)$ that

$$ w(u,v) \cdot \frac{\alpha \cdot \log n}{d_u^{\text{out}} \cdot \lambda_2} < 1, \qquad w(u,v) \cdot \frac{\alpha \cdot \log n}{d_v^{\text{in}} \cdot \lambda_2} < 1. $$

Moreover, the expected number of edges in $H$ equals to

$$\sum_{e=(u,v)} p_e \leqslant \sum_{e=(u,v)} p_u(u,v) + p_v(u,v) = \frac{\alpha \cdot \log n}{\lambda_2} \sum_{e=(u,v)} \left( \frac{w(u,v)}{d_u^{\mathrm{out}}} + \frac{w(u,v)}{d_v^{\mathrm{in}}} \right)$$

$$= O\left( \frac{n \log n}{\lambda_2} \right),$$

and thus by Markov's inequality we have that with constant probability the number of sampled edges $|F| = O\left((1/\lambda_2) \cdot n \log n\right)$.

*Proof of $\theta_k(H) = \Omega(\theta_k(G))$.* Next we show that the sparsified graph constructed by the algorithm preserves $\theta_k(G)$ up to a constant factor. Without loss of generality, let $S_0, \ldots, S_{k-1}$ be the optimal $k$ clusters such that

$$\Phi_G(S_0, \ldots, S_{k-1}) = \theta_k(G).$$

For any edge $e = (u,v)$ satisfying $u \in S_j$ and $v \in S_{j-1}$ for some $1 \leqslant j \leqslant k - 1$, we define a random variable $Y_e$ by

$$Y_e = \begin{cases} w(u,v)/p_e & \text{with probability } p_e, \\ 0 & \text{otherwise.} \end{cases}$$

We also define random variables $Z_1, \ldots, Z_{k-1}$, where $Z_j$ $(1 \leqslant j \leqslant k-1)$ is defined by

$$Z_j = \sum_{\substack{e=\{u,v\}\in E[G] \\ u\in S_j, v\in S_{j-1}}} Y_e.$$

By definition, we have that

$$\mathbb{E}[Z_j] = \sum_{\substack{e=\{u,v\}\in E[G] \\ u\in S_j, v\in S_{j-1}}} \mathbb{E}[Y_e] = \sum_{\substack{e=\{u,v\}\in E[G] \\ u\in S_j, v\in S_{j-1}}} w(u,v) = w(S_j, S_{j-1}).$$

Moreover, we look at the second moment and have that

$$\sum_{\substack{e=\{u,v\}\in E[G] \\ u\in S_j, v\in S_{j-1}}} \mathbb{E}\left[Y_e^2\right] = \sum_{\substack{e=\{u,v\}\in E[G] \\ u\in S_j, v\in S_{j-1}}} p_e \cdot \left( \frac{w(u,v)}{p_e} \right)^2$$

$$= \sum_{\substack{e=\{u,v\}\in E[G] \\ u\in S_j, v\in S_{j-1}}} \frac{(w(u,v))^2}{p_e}$$

$$\leqslant \sum_{\substack{e=\{u,v\}\in E[G] \\ u\in S_j, v\in S_{j-1}}} \frac{(w(u,v))^2}{w(u,v)} \cdot \frac{\lambda_2 \cdot d_u^{\mathrm{out}}}{\alpha \log n}$$

$$= \frac{\lambda_2}{\alpha \log n} \cdot \sum_{\substack{e=\{u,v\}\in E[G] \\ u\in S_j, v\in S_{j-1}}} w(u,v) \cdot d_u^{\mathrm{out}}$$

$$\leqslant \frac{\lambda_2}{\alpha \log n} \cdot \Delta_j^{\mathrm{out}} \cdot w(S_j, S_{j-1}),$$

where $\Delta_j^{\mathrm{out}}$ is the maximum of the out degree of vertices in $S_j$ and the first inequality follows by the fact that

$$p_e = p_u(u,v) + p_v(u,v) - p_u(u,v)p_v(u,v) \geqslant p_u(u,v) = w(u,v) \cdot \frac{\alpha \log n}{\lambda_2 \cdot d_u^{\mathrm{out}}}.$$

In addition, it holds for any $e = (u,v), u \in S_j, v \in S_{j-1}$ that

$$\left| \frac{w(u,v)}{p_e} \right| \leqslant \left| \frac{w(u,v)}{p_u(u,v)} \right| \leqslant \frac{\lambda_2 \cdot \Delta_j^{\mathrm{out}}}{\alpha \cdot \log n}.$$

We apply Bernstein's Inequality (Lemma B.2), and obtain for any $1 \leqslant j \leqslant k-1$ that

$$
\begin{aligned}
&\mathbb{P}\left[|Z_j - w(S_j, S_{j-1})| \geqslant (1/2) \cdot w(S_j, S_{j-1})\right] \\
&= \mathbb{P}\left[|Z_j - \mathbb{E}[Z_j]| \geqslant (1/2) \cdot \mathbb{E}[Z_j]\right] \\
&\leqslant 2 \cdot \exp\left(-\frac{\mathbb{E}[Z_j]^2/4}{2\left(\frac{\lambda_2}{\alpha \log n} \cdot \Delta_j^{\mathrm{out}} \cdot w(S_j, S_{j-1}) + \frac{\lambda_2 \cdot \Delta_j^{\mathrm{out}}}{\alpha \cdot \log n} \cdot \frac{1}{6} \cdot w(S_j, S_{j-1})\right)}\right) \\
&\leqslant 2 \cdot \exp\left(-\frac{\alpha \cdot \log n \cdot \mathbb{E}[Z_j]}{10 \cdot \lambda_2 \cdot \Delta_j^{\mathrm{out}}}\right).
\end{aligned}
$$

Hence, with high probability cut values $w(S_j, S_{j-1})$ for all $1 \leqslant j \leqslant k-1$ are approximated up to a constant factor. Using the same technique, we can show that with high probability the volumes of all the sets $S_0, \ldots, S_{k-1}$ are approximately preserved in $H$ as well. Combining this with the definition of $\Phi$, we have that $\Phi_G(S_0, \ldots, S_{k-1})$ and $\Phi_H(S_0, \ldots, S_{k-1})$ are approximately the same up to a constant factor. Since $S_0, \ldots, S_{k-1}$ are the sets that maximising the value of $\theta_k(G)$, we have that $\theta_k(H) = \Omega(\theta_k(G))$.

*Proof of $\lambda_2(\mathcal{L}_H) = \Omega(\lambda_2(\mathcal{L}_G))$.* Finally, we prove that the top $n-1$ eigenspace is approximately preserved in $H$. Let $\overline{\mathcal{L}}_G$ be the projection of $\mathcal{L}_G$ on its top $n-1$ eigenspaces. We can write $\overline{\mathcal{L}}_G$ as

$$
\overline{\mathcal{L}}_G = \sum_{i=2}^{n} \lambda_i f_i f_i^*.
$$

With a slight abuse of notation we call $\overline{\mathcal{L}}_G^{-1/2}$ the square root of the pseudoinverse of $\overline{\mathcal{L}}_G$, i.e.,

$$
\overline{\mathcal{L}}_G^{-1/2} = \sum_{i=2}^{n} (\lambda_i)^{-1/2} f_i f_i^*.
$$

We call $\overline{\mathcal{I}}$ the projection on $\mathrm{span}\{f_2, \ldots, f_n\}$, i.e.,

$$
\overline{\mathcal{I}} = \sum_{i=2}^{n} f_i f_i^*.
$$

We will prove that the top $n-1$ eigenspaces of $\mathcal{L}_G$ are preserved. To prove this, recall that the probability of any edge $e = (u, v)$ being sampled in $H$ is

$$
p_e = p_u(u, v) + p_v(u, v) - p_u(u, v) \cdot p_v(u, v),
$$

and it holds that $\frac{1}{2}(p_u(u, v) + p_v(u, v)) \leqslant p_e \leqslant p_u(u, v) + p_v(u, v)$. Now for each edge $e = (u, v)$ of $G$ we define a random matrix $X_e \in \mathbb{C}^{n \times n}$ by

$$
X_e = \begin{cases} w_H(u, v) \cdot \overline{\mathcal{L}}_G^{-1/2} b_e b_e^* \overline{\mathcal{L}}_G^{-1/2} & \text{if } e = (u, v) \text{ is sampled by the algorithm,} \\ 0 & \text{otherwise,} \end{cases}
$$

where the vector $b_e$ is defined by $b_e = \left(\omega_{2\lceil 2\pi \cdot k\rceil}\chi_u - \omega_{2\lceil 2\pi \cdot k\rceil}^*\chi_v\right)$ and for any vertex $u$ the normalised indicator vector $\chi_u$ is defined by $\chi_u(u) = 1/\sqrt{d_u}$, and $\chi_u(v) = 0$ for any $v \neq u$. Notice that

$$
\sum_{e \in E[G]} X_e = \sum_{\text{sampled edges } e=(u,v)} w_H(u, v) \cdot \overline{\mathcal{L}}_G^{-1/2} b_e b_e^* \overline{\mathcal{L}}_G^{-1/2} = \overline{\mathcal{L}}_G^{-1/2} \mathcal{L}_H' \overline{\mathcal{L}}_G^{-1/2},
$$

where it follows by definition that

$$
\mathcal{L}_H' = \sum_{\text{sampled edges } e=(u,v)} w_H(u, v) \cdot b_e b_e^*
$$

is essentially the Laplacian matrix of $H$ but is normalised with respect to the degrees of the vertices in the original graph $G$, i.e., $\mathcal{L}_H' = D_G^{-1}D_H - D_G^{-1/2}A_H D_G^{-1/2}$. We will prove that, with high

probability, the top $n-1$ eigenspaces of $\mathcal{L}'_H$ and $\mathcal{L}_G$ are approximately the same. Later we will show the same holds for $\mathcal{L}_H$ and $\mathcal{L}'_H$, which implies that $\lambda_2(\mathcal{L}'_H) = \Omega(\lambda_2(\mathcal{L}_G))$.

We will use the matrix Chernoff bound for our proof. We start looking at the first moment of the expression above:

$$
\mathbb{E}\left[\sum_{e\in E} X_e\right] = \sum_{e=(u,v)\in E[G]} p_e \cdot w_H(u,v) \cdot \overline{\mathcal{L}}_G^{-1/2} b_e b_e^* \overline{\mathcal{L}}_G^{-1/2}
$$

$$
= \sum_{e=(u,v)\in E[G]} p_e \cdot \frac{w(u,v)}{p_e} \cdot \overline{\mathcal{L}}_G^{-1/2} b_e b_e^* \overline{\mathcal{L}}_G^{-1/2}
$$

$$
= \overline{\mathcal{L}}_G^{-1/2} \mathcal{L}_G \overline{\mathcal{L}}_G^{-1/2} = \overline{\mathcal{I}}.
$$

Moreover, for any sampled $e = (u,v) \in E$ we have that

$$
\|X_e\| \leqslant w_H(u,v) \cdot b_e^* \overline{\mathcal{L}}_G^{-1/2} \overline{\mathcal{L}}_G^{-1/2} b_e = \frac{w(u,v)}{p_e} \cdot b_e^* \overline{\mathcal{L}}_G^{-1} b_e \leqslant \frac{w(u,v)}{p_e} \cdot \frac{1}{\lambda_2} \cdot \|b_e\|^2
$$

$$
\leqslant \frac{2\lambda_2}{\alpha \cdot \log n \cdot \left(\frac{1}{d_u^{\text{out}}} + \frac{1}{d_v^{\text{in}}}\right)} \cdot \frac{1}{\lambda_2}\left(\frac{1}{d_u^{\text{out}}} + \frac{1}{d_v^{\text{in}}}\right) \leqslant \frac{2}{\alpha \log n},
$$

where the second inequality follows by the min-max theorem of eigenvalues. Now we apply the matrix Chernoff bound (Lemma B.3) to analyse the eigenvalues of $\sum_{e\in E} X_e$, and build a connection between $\lambda_2(\mathcal{L}'_H)$ and $\lambda_2(\mathcal{L}_G)$. By setting the parameters of Lemma B.3 by $\mu_{\max} = \lambda_{\max}\left(\mathbb{E}\left[\sum_{e\in E[G]} X_e\right]\right) = \lambda_{\max}\left(\overline{\mathcal{I}}\right) = 1$, $R = 2/(\alpha \cdot \log n)$ and $\delta = 1/2$, we have that

$$
\mathbb{P}\left[\lambda_{\max}\left(\sum_{e\in E[G]} X_e\right) \geqslant 3/2\right] \leqslant n \cdot \left(\frac{e^{1/2}}{(1+1/2)^{3/2}}\right)^{\alpha\log n/2} = O\left(1/n^c\right)
$$

for some constant $c$. This gives us that

$$
\mathbb{P}\left[\lambda_{\max}\left(\sum_{e\in E[G]} X_e\right) \leqslant 3/2\right] = 1 - O(1/n^c). \tag{12}
$$

On the other side, since our goal is to analyse $\lambda_2(\mathcal{L}'_H)$ with respect to $\lambda_2(\mathcal{L}_G)$, it suffices to work with the top $(n-1)$ eigenspace of $\mathcal{L}_G$. Since $\mathbb{E}\left[\sum_{e\in E} X_e\right] = \overline{\mathcal{I}}$, we can assume without loss of generality that $\mu_{\min} = 1$. Hence, by setting $R = 2/(\alpha \cdot \log n)$ and $\delta = 1/2$, we have that

$$
\mathbb{P}\left[\lambda_{\min}\left(\sum_{e\in E[G]} X_e\right) \leqslant 1/2\right] = n \cdot \left(\frac{e^{-1/2}}{(1/2)^{1/2}}\right)^{\alpha\log n/2} = O\left(1/n^c\right)
$$

for some constant $c$. This gives us that

$$
\mathbb{P}\left[\lambda_{\min}\left(\sum_{e\in E[G]} X_e\right) > 1/2\right] = 1 - O(1/n^c). \tag{13}
$$

Combining (12), (13), and the fact of $\sum_{e\in E[G]} X_e = \overline{\mathcal{L}}_G^{-1/2} \mathcal{L}'_H \overline{\mathcal{L}}_G^{-1/2}$, with probability $1 - O\left(1/n^c\right)$ it holds for any non-zero $x \in \mathbb{C}^n$ in the space spanned by $f_2,\ldots,f_n$ that

$$
\frac{x^* \overline{\mathcal{L}}_G^{-1/2} \mathcal{L}'_H \overline{\mathcal{L}}_G^{-1/2} x}{x^* x} \in (1/2, 3/2). \tag{14}
$$

By setting $y = \overline{\mathcal{L}}_G^{-1/2} x$, we can rewrite (14) as

$$
\frac{y^* \mathcal{L}'_H y}{y^* \overline{\mathcal{L}}_G^{1/2} \overline{\mathcal{L}}_G^{1/2} y} = \frac{y^* \mathcal{L}'_H y}{y^* \overline{\mathcal{L}}_G y} = \frac{y^* \mathcal{L}'_H y}{y^* y}\frac{y^* y}{y^* \overline{\mathcal{L}}_G y} \in (1/2, 3/2).
$$

Since $\dim(\mathrm{span}\{f_2, \ldots, f_n\}) = n - 1$, we have just proved there exist $n - 1$ orthogonal vectors whose Rayleigh quotient with respect to $\mathcal{L}'_H$ is $\Omega(\lambda_2(\mathcal{L}_G))$. By the Courant-Fischer Theorem, we have

$$\lambda_2(\mathcal{L}'_H) \geqslant \frac{1}{2} \lambda_2(\mathcal{L}_G). \tag{15}$$

It remains to show that $\lambda_2(\mathcal{L}_H) = \Omega\left(\lambda_2(\mathcal{L}'_H)\right)$, which implies that $\lambda_2(\mathcal{L}_H) = \Omega\left(\lambda_2(\mathcal{L}_G)\right)$ by (15). By the definition of $\mathcal{L}'_H$, we have that for the Laplacian $\mathcal{L}_H = D_H^{-1/2} D_G^{1/2} \mathcal{L}'_H D_G^{1/2} D_H^{-1/2}$. Therefore, for any $x \in \mathbb{C}^n$ and $y = D_G^{1/2} D_H^{-1/2} x$, it holds that

$$\frac{x^* \mathcal{L}_H x}{x^* x} = \frac{y^* \mathcal{L}'_H y}{x^* x} \geqslant \frac{1}{2} \cdot \frac{y^* \mathcal{L}'_H y}{y^* y}, \tag{16}$$

where the last equality follows from the fact that the degrees in $H$ and $G$ differ just by a constant multiplicative factor, and therefore,

$$y^* y = \left(D_G^{1/2} D_H^{-1/2} x\right)^* \left(D_G^{1/2} D_H^{-1/2} x\right) = x^* D_G D_H^{-1} x \geqslant \frac{1}{2} \cdot x^* x.$$

Finally, we show that (16) implies that $\lambda_2(\mathcal{L}_H) \geqslant (1/2) \cdot \lambda_2(\mathcal{L}'_H)$. To see this, let $S_1 \subseteq \mathbb{C}^n$ be a (2)-dimensional subspace of $\mathbb{C}^n$ such that

$$\lambda_2(\mathcal{L}_H) = \max_{x \in S_1} \frac{x^* \mathcal{L}_H x}{x^* x}.$$

Let $S_2 = \left\{ D_G^{1/2} D_H^{-1/2} x : x \in S_1 \right\}$. Notice that since $D_G^{1/2} D^{-1/2}$ is full rank, $S_2$ has dimension 2. Therefore,

$$\lambda_2(\mathcal{L}'_H) = \min_{S: \, \dim(S) = 2} \max_{y \in S} \frac{y^* \mathcal{L}'_H y}{y^* y} \leqslant \max_{y \in S_2} \frac{y^* \mathcal{L}'_H y}{y^* y} \leqslant 2 \max_{x \in S_1} \frac{x^* \mathcal{L}_H x}{x^* x} = 2 \lambda_2(\mathcal{L}_H), \tag{17}$$

where the last inequality follows by (16). Combining (15) with (17) gives us that $\lambda_2(\mathcal{L}_H) = \Omega(\lambda_2(G))$. This concludes the proof. $\square$

## C  Omitted details from Section 5

### C.1  UN Comtrade Data Preparation

The API provided by the UN gives a lot of flexibility on the type of selected data. It is possible to specify the *product type* to either trade in goods (e.g., oil, wood, and appliances) or services (e.g., financial services, and construction services). Moreover, the *classification code* can be selected, which we set to the Harmonised System (HS). The HS categorises goods according to a 6-digit classification code (e.g., 060240, where the first two digits "06" represents "plants", the second two digits "02" represents "alive", and the last two digits "40" code for "roses"). The *reporting* countries and *partner* countries can also be specified, where the reporting country reports about its own reported tradeflow with partner countries. The settings we used to download the data for our experiments were Goods on an annual frequency, the HS code as reported, over the period from 2002 to 2017, with all reporting and all partner countries, all trade flows and all HS commodity codes. The total size of the data in zipped files is 99.8GB, where each csv file (for every year) contains around $20,000,000$ lines.

For every pair of countries $j$ and $\ell$, where $j$ is the reporting country and $\ell$ is the partner country, the database contains the amount that country $j$ imports from country $\ell$ for a specific commodity, and also the amount $j$ exports to $\ell$. There are several cases where countries $j$ and $\ell$ report different trading amounts with each other. Usually, the larger value is considered more accurate and is used instead of the average [12]. To construct the digraph of the world trade network and its corresponding adjacency matrix, we fill in each entry of the adjacency matrix $M^c$ for commodity $c$ as follows: for each pair of countries $j$ and $\ell$, we compute $d_{j\ell}^c = e_{j\ell}^c - e_{\ell j}^c$, where $e_{j\ell}^c$ is the amount country $j$ exports to country $\ell$ for commodity $c$. If $d_{j\ell}^c > 0$, we set $M_{j\ell}^c = d_{j\ell}^c$ and $M_{\ell j}^c = 0$. If $d_{j\ell}^c < 0$ (and thus $d_{\ell j}^c > 0$), we set $M_{\ell j}^c = d_{\ell j}^c$ and $M_{j\ell}^c = 0$.

For our experiments we investigate the trade in "Mineral Fuels, mineral oils, and products of their distillation" (HS code 27), and the trade in "Wood and articles of wood" (HS code 44).

Figure 6: Change in clustering of the IOTN over period 2006–2009 with $k = 4$ using DD-SYM method. Red countries form the start of the trade chain, and yellow countries the end of the trade chain. Countries coloured white have no data.

## C.2 DD-SYM Plots International Oil Trade

We plot the cluster visualisations for the DD-SYM algorithm in Figure 6 on the international oil trade network, over the period 2006-2009. The clusters between 2006 and 2007 are almost identical, and then there is a shift in the clustering structure between 2007 and 2008. This change occurs one year before the change in the SimpleHerm method, and this change is also one year earlier than the changes found in the complex network analysis literature [1, 33]. This indicates that the SimpleHerm clustering result is more in line with other literature.

## C.3 International Wood Trade

For comparison we visualise the clustering result of the DD-SYM method over the period of 2006 – 2009, see Figure 7. In addition, Figure 8 compares the symmetric difference of the clusters returned by different algorithms over the consecutive years. Again, we notice that our algorithm finds a peak around the economic crisis of 2008, and another peak is found between 2005 and 2006. We could not find any literature reasoning about the peak between 2005 and 2006, but it would be interesting to analyse this further. The symmetric difference returned by the DD-SYM method is more noisy.

## C.4 Results on Data Science for COVID-19 Dataset

The *Data Science for COVID-19 Dataset* (DS4C) [19] contains information about $3519$ South Korean COVID-19 cases, and we use directed edges to represent how the virus is transmitted among the individuals. We notice that there are only $831$ edges in the graph and there are many connected components of size $2$. To take this into account, we run our algorithm on the largest connected component of the infection graph, which consists of $67$ vertices and $66$ edges. Applying the complex-valued Hermitian matrix and the eigenvector associated with the smallest eigenvalue, the spectral embedding is visualised in Figure 9.

We notice several interesting facts. First of all, we do not see all the individual nodes of the graph in this embedding. This is because many embedded points are overlapped, which happens if they have the same in and outgoing edges. Moreover, from cluster $S_0$ to $S_1$ there is 1 edge, from $S_1$ to $S_2$ there are 51 edges and from $S_2$ to $S_3$ there are 5 edges. That means there are $1 + 51 + 5 = 57$ edges that lie along the path, out of 66 edges in total. This concludes that our algorithm has successfully clustered the vertices such that there is a large flow ratio along the clusters.

Figure 7: Change in clustering of the IWTN over period 2006-2009 with $k = 4$ using DD-SYM method. Red countries form the start of the trade chain, and yellow countries the end of the trade chain. Countries coloured white have no data.

Figure 8: Comparison of the symmetric difference of the returned clusters between two consecutive years on the IWTN.

Secondly, due to the limited size of the dataset, it is difficult for us to draw a more significant conclusion from the experiment. However, we do notice that the cluster $S_1$ actually consists of one individual: a super spreader. This individual infected $51$ people in cluster $S_2$. We believe that, with the development of many tracing Apps across the world and more data available in the near future, our algorithm could become a useful tool for disease tracking and policy making.

Figure 9: Clustering output on the largest connected component of the DS4C dataset, where $k = 4$. Clusters are labelled according to their position in the ordering that maximises the flow ratio.