[Reviews · NeurIPS 2020]

Review 1

Summary and Contributions: The paper considers a graph clustering on directed graphs. The authors introduce a new notion of clustering objective denoted by flow ratio. For any ordered partition of vertex set V into k pairwise disjoint subset (S0, ..., Sk-1), the flow ratio of the partition is sum of the average flow (i.e. w(i, i+1)/Vol(Si)+Vol(Si+1)) along the path from S0 to Sk-1. The optimal clustering is the partitioning of V that maximizes the flow ratio among all possible partitions. The authors represent the directed graph using the Hermitian adjacency matrix. The main contribution of this work is proving that the optimal clustering is well-embedded with the first eigenvector of the corresponding Laplacian matrix. Equipped with this observation, authors design an algorithm to find clustering. The algorithm is very similar to the standard spectral clustering algorithms for undirected graph; It first estimates the embedding and then runs k-means on the corresponding embedding. They show that for graphs with few clusters (i.e., poly(log n)) the algorithm runs in nearly linear time. Moreover, they prove that the clustering returned by the algorithm has a small symmetric difference with optimal clustering where the recovery quality depends on the gap between the first and second eigenvalues of the Laplacian. Finally, the authors show how to improve the runtime of the algorithm by subsampling edges with probabilities proportional to the weight by degree. In addition, they evaluate the performance of their algorithm on the stochastic block model and UN Comtrade Dataset.

Strengths: This paper has a conceptual contribution in considering the flow ratio notion for clustering directed graphs. It is an initial step towards applying known spectral techniques for undirected graphs to the directed graphs with the help of complex Hermitian adjacency matrix. Although the algorithm and most of the theoretical analysis are very similar to the standard spectral clustering literature such as work by Peng, Sun and Zanetti, the novel observation of this work is embedding the points into the space of dimension 2 with the help of complex Hermitian matrix. This approach shows how to differentiate points from different clusters using angles, although to find the clustering for undirected graphs it is crucial to look at the subspace of dimension k that is spanned by the first k eigenvectors.

Weaknesses: The authors succeed in showing that the standard spectral techniques can be applied to the directed graph clustering using flow ratio formulation. However, the notion of flow ratio is not well-motivated. It is not clear why one should consider a specific path between clusters. In other words, the relationship between pairwise different clusters is not captured in the objective function and it only takes care of consecutive terms which is a restrictive definition in my opinion. Moreover, since this is a new objective function for directed graph clustering the authors should justify the hardness of finding the optimal clustering, also they should compare it with other clustering objectives for directed graphs. Regarding the experiments on stochastic block model, first, it's not clear how large the input graph is, also, the number of clusters (i.e, k) is a small constant, so it's not clear from the experiments if the sublinear algorithm scales very well on large graphs. Second, the number of different runs in Figure 1 seems to be very small which might not be enough to get a good concentration in the accuracy of the results. Finally, the authors evaluate the quality of their methods using the Adjusted Rand Index. It might be useful to report the quality of the generated output using other known evaluation methods for spectral clusterings such as inner and outer conductance, precision and recall.

Correctness: Yes both theoretical claims and empirical methodology are correct. In line 185 of the main file it's not clear for me why do you have (1+APT)(yk-1)? It shouldn't be APT*(yk-1)?

Clarity: The paper is well-written, and the algorithms and proofs are well-explained.

Relation to Prior Work: The authors clearly explain their theoretical contribution comparing to the previous works.

Reproducibility: Yes

Additional Feedback:


Review 2

Summary and Contributions: After reading the authors' rebuttal, my evaluation remains the same. The work studies clustering of directed graphs by spectral means. Traditional spectral clustering on undirected graphs seek to find dense clusters that induce small cuts. Instead here the goal is to find a directed structure, for example, a cluster that has mostly outgoing edges and one that has mostly incoming edges (so sparse clusters inducing large cuts are welcome). The authors introduce a notion of flow ratio and discuss algorithms to find a clustering that approximately maximize that ratio. The paper contains experiments where the aforementioned algorithms find patterns in international trade markets (e.g. clusters of crude oil exporting countries).

Strengths: The topic is interesting and matches the NeurIPS audience. The techniques are nontrivial and yield interesting insights. The whole analysis and the algorithms seem theoretically well grounded. The spectral analysis is of interest on its own (I remark that in many points it bears similarities with previous work). Thus I believe the work connects nicely the original goal of clustering with the spectral analysis for directed graphs and with the resulting approximation algorithms. The experiments are interesting as well, and suggest that the proposed algorithm is indeed capable of finding the cluter structure that one would expect. In particular the authors recover the crude oil international market structure (say, exporters vs importers) using data involving 246 different countries and regions over several years.

Weaknesses: The main weakness of the paper is that here and there the exposition is poor. In particular, it is hard to make the necessary connections and understand where the discussion is going. There are several technical results stated in a "standalone" form although they are clearly meant to be connected/used somewhere else. This makes it quite hard to understand the key ideas of the paper and in what they differ from previous work. In particular, many concepts and results are similar to [22] (Peng et al @ SICOMP 17).

Correctness: I could not verify the proofs of the claims although the results make sense. There is at least one mathematical expression that I found suspicious since it is undefined (division by 0) when k=2 and G has a single edge or is bipartite (see detailed comments below). I do not know if this is only a corner case.

Clarity: The paper is fairly well written, but as I said it is generally hard to understand where the discussion is heading. There are also a few sentences that are confusing or incorrect (for example, at the very beginning of the introduction the paper presents a complicate expression and none of the symbols involved were defined before). I think the presentation can be improved.

Relation to Prior Work: Yes and no. The authors acknowledge previous work in the sense that they correctly point to the relevant papers. What leaves me in doubt is that it is not clear what is similar to what. For example, the authors introduce in Equation (3) a kind of indicator vector for the clusters. Similar things are standard, so this is an (interesting) variation, but this it is not written. (Specifically, the equation just above (3) is the standard normalized indicator vector, and (3) does a linear combination which is the novelty part). This leaves the non-specialist reader like me in doubt; what is conceptually new, and what is just an application of standard techniques?

Reproducibility: Yes

Additional Feedback: Detailed comments: - line 42: "any set of vertices" is confusing, here there is a partition - lines 43-35: I could not understand this expression as none of the involved symbols has been introduced except S_j - lines 73-74: "in particular we have vol(G) = vol(V) = 2m" this is not true since you define d() as weights instead of degrees - lines 98-99: "the fraction of directed edges with endpoints in S_j or S_{j-1} which cross the cut from S_j and S_{j-1}" this sounds like the direction of the edge is irrelevant (it is also oddly phrased) - line 122: this expression seems to have a 0 denominator, for example when k=2 and G is a single edge; this is strange - lines 138-139: "we propose to apply k-means on the embedded points from f_1" I do not understand this sentence, in particular "from f_1" and what points the sentence is talking about - S 4.2: I cannot understand how the three lemmata imply the theorem; I see they are related, but they are just listed without explaining the connection


Review 3

Summary and Contributions: This paper studies k-way partitions of directed graphs using k (complex) eigenvectors of the directed Laplacian. It gives bounds similar to those for k-way partitions of undirected graphs, and demonstrates good performances of the algorithms experimentally on small synthetic and real world graphs.

Strengths: Extending spectral clustering into directed graphs is a difficult task: it's quite surprising to me that spectral techniques and their guarantees can be readily extended to directed graphs. The experimental verifications of the algorithms are fairly through.

Weaknesses: Aside from the extension to the complex domain, it's not clear what are the new ideas in this paper compared to spectral methods for (k-way) partitioning undirected graphs. The experiments were mostly for small sized graphs, and it's not clear how effective the algorithm would scale to larger instances. Even the largest dataset for oil trading only involves < 300 vertices (but with about 100 commodities): I suspect the data becomes much smaller after the input is converted to graph form. If the data size is indeed large, it would be useful to directly mention the sizes of the intermediate graphs generated.

Correctness: I believe the algorithm and the bounds claimed. The experimental set up is sound, although I'd like to have seen some comparisons with other methods such as pagerank or matrix factorization based methods. While those methods are not geared toward the flow ratios studied here, I believe their output clusters can still be compared directly.

Clarity: The paper introduces the problem and methods clearly, but I find some of the formal components a bit lacking: for example, k was assumed to be O(log^{c}n) on line 144 and omitted from subsequent bounds involving running times. A pseudo-code of the overall algorithm is also not present in sec 4: this made it difficult to figure out the overall algorithmic picture.

Relation to Prior Work: Prior works, especially for spectral partitioning of directed graphs, are discussed systematically.

Reproducibility: Yes

Additional Feedback: I believe the paper has some interesting ideas, but have difficulty figuring out its improvements over previous results. I feel there are enough theoretical new ideas for the paper to be accepted, although also feel that the ideas are still an iteration away from being directly used on large networks.

[Author Response · NeurIPS 2020]

We thank all reviewers for their detailed and insightful reports. As pointed out by most reviewers, the key to our work is that, by using complex-valued matrices, the $k$ clusters in a digraph can be well separated by *angles* in $\mathbb{R}^2$ with the help of a *single* eigenvector. The result in Peng et al. [22] only holds for undirected graphs and $k$ eigenvectors are needed to construct an embedding. Generalising their result into the setting of digraphs is a mathematically involved task.

**Reply to R1:** We thank R1 for recognising the conceptual contribution of introducing the flow ratio. For the comment on pairwise different clusters not being captured in the objective function, notice that if one takes all the pairwise clusters into account, the objective function would involve $\Theta(k^2)$ terms. Even if the remaining $\Theta(k^2)$ terms are much smaller than the ones along the path, their sum could still be dominant, leaving little valuable information on the cluster structure for the objective function. Instead, we show that when the relationship between clusters present a structure, one should *only* take the cut values along the path into account. This difference makes it difficult to compare our objective function with others. The motivation for studying a flow ratio is witnessed by our experiments on the international trade data set and South Korean COVID-19 dataset, on which meaningful clusters are found. On the problem's hardness, notice that, when $k = 2$, the optimal solution is the bipartition of $V$ that maximises $w(S_0, S_1)$. This is exactly the MAX DICUT problem, which is NP-hard (cf. Goemans and Williamson, JACM'95). We will add this in the final version.

For the DSBM experiments, the sizes of the input graphs are $n = 1000$ for Fig.1, and $n = 2000$ for Fig.2. We accidentally omitted these, but will report these values in the final version. The number of vertices $n$ and clusters $k$ we choose are in line with many references for (D)SBM experiments, and we do remark that the results of different runs are very close. Hence, we believe that the number of runs needed for good convergence should not play a significant role here so that using 5 runs is sufficient. We follow reference [9] and use the ARI value for evaluating the clustering quality. We will consider reporting other metrics such as the number of misclassified vertices, precision or recall. The inner and outer conductance are not suitable for our setting since cluster structures in Fig.1 are characterised by edge directions instead of densities. R1 is correct that the term in L.185 should be $\mathsf{APT}/(\gamma_k(G) - 1)$, and this will be corrected.

**Reply to R2:** We thank the reviewer for highlighting both the new theoretical and experimental contributions of the paper, and appreciate their comments on the paper exposition. The page limit restricted extended discussion, but we will improve the presentation. Regarding Eq.(3), we did not add more discussions/references since both (3) and the equation before are new. In particular, the equation before (3) already uses angles to differentiate clusters, so it is not a standard normalised indicator vector as used for undirected graphs. Because both the equations are conceptually new, we did not add more discussions here. Instead, L.110-115 compared these two equations with previous work.

Regarding the detailed comments: For L.42, we do mention that "for any set of vertices $S_0, \ldots, S_{k-1}$ *that forms a partition*"; For L.43-45, we will move the definition of $w(\cdot, \cdot)$ and $\mathrm{vol}(\cdot)$ up; For L.73-74, we will drop the second equality; For L.98-98, we will rewrite it as "which cross the cut from $S_j$ *to* $S_{j-1}$"; For L.138-139, here we use $f_1$ as an embedding, and every vertex $v$ maps to $f_1(v) \in \mathbb{C}$; these $n$ embedded points are the input of $k$-means: we will make it more clear; On the comment about S4.2, the three lemmas are used as key components to prove the theorem. Specifically, we prove by contradiction through a quite technical analysis that if the theorem does not hold, then one of the lemmas does not hold. We will add some discussion about this in the final version. Now we address R2's concern on L.112 (and Correctness). Notice that the denominator is equal to 0 if $\sum_{j=1}^{k-1} w(S_j, S_{j-1})/k(\mathrm{vol}(S_j) + \mathrm{vol}(S_{j-1})) = 1/4$. This summation is maximised if the graph is complete $k$-partite and every partition is a cluster. Moreover, the case in which $\theta_k$ achieves maximum occurs when $k = 2$, which leads to $1 - (4/k) \cdot \theta_k = 0$. This is indeed a corner case, we will clarify this (or adjust constants in the denominator of $\gamma_k(G)$ to avoid this). We thank the reviewer for pointing this out.

**Reply to R3**: We thank the reviewer for their thoughtful comments. Regarding the comment on comparing our method with the ones for partitioning undirected graphs, notice that our key contribution is to introduce the flow ratio, and demonstrate that, thanks to complex-valued matrices, the cluster structure can be encoded into *a single eigenvector* (instead of $k$ eigenvectors); different clusters are then separated by *angles* in the $\mathbb{R}^2$ embedding. R1 highlighted this as a strength. For the experiments, the DSBM graphs have 1K to 2K vertices, similar in size to other references. We also evaluated our algorithm on the UN Comtrade Dataset (zipped 99.8GB). Therefore, we do believe our algorithm has been tested on large and relevant graph instances.

Regarding R3's comment on comparison to PageRank and matrix factorisation methods, we specifically compared our algorithm with all the previous ones addressing the same goal. To the best of our knowledge, there is no PageRank-based algorithm to find clusters in digraphs whose structure are defined with respect to flow imbalance; furthermore, all matrix multiplication based algorithms have a runtime of $\Omega(n^\omega)$, which is computationally more expensive than our algorithm.

Regarding the comments on the assumption on $k$, notice that for most applications $k$ is small and it is reasonable to focus on $k = O(\mathrm{poly} \log n)$. The use of $\tilde{O}$-notation to hide $\mathrm{poly} \log n$ factors is a standard practice in studying nearly-linear time graph algorithms. About the pseudocode: because of the simplicity of the algorithm, whose entire description is included in L.137-142 and L.154-164, we did not include the pseudocode due to space limitations. Should R3 strongly believe that inclusion of pseudocode would significantly improve the readability of the paper, we can aim to do so.

[Meta-Review · NeurIPS 2020]

All the reviews are positive. I also agree that this paper should be part of the program.